# Major genetic discontinuity and novel toxigenic species in *Clostridioides difficile* taxonomy

Daniel R Knight[1,2]*, Korakrit Imwattana[2,3], Brian Kullin[4], Enzo Guerrero-Araya[5,6], Daniel Paredes-Sabja[5,6,7], Xavier Didelot[8], Kate E Dingle[9], David W Eyre[10], César Rodríguez[11], Thomas V Riley[1,2,12,13]*

[1]Medical, Molecular and Forensic Sciences, Murdoch University, Murdoch, Australia; [2]School of Biomedical Sciences, the University of Western Australia, Nedlands, Australia; [3]Department of Microbiology, Faculty of Medicine Siriraj Hospital, Mahidol University, Bangkok, Thailand; [4]Department of Pathology, University of Cape Town, Cape Town, South Africa; [5]Microbiota-Host Interactions and Clostridia Research Group, Facultad de Ciencias de la Vida, Universidad Andrés Bello, Santiago, Chile; [6]Millenium Nucleus in the Biology of Intestinal Microbiota, Santiago, Chile; [7]Department of Biology, Texas A&M University, College Station, United States; [8]School of Life Sciences and Department of Statistics, University of Warwick, Coventry, United Kingdom; [9]Nuffield Department of Clinical Medicine, University of Oxford, National Institute for Health Research (NIHR) Oxford Biomedical Research Centre, John Radcliffe Hospital, Oxford, United Kingdom; [10]Big Data Institute, Nuffield Department of Population Health, University of Oxford, National Institute for Health Research (NIHR) Oxford Biomedical Research Centre, John Radcliffe Hospital, Oxford, United Kingdom; [11]Facultad de Microbiología & Centro de Investigación en Enfermedades Tropicales (CIET), Universidad de Costa Rica, San José, Costa Rica; [12]Department of Microbiology, PathWest Laboratory Medicine, Queen Elizabeth II Medical Centre, Nedlands, Australia; [13]School of Medical and Health Sciences, Edith Cowan University, Joondalup, Australia

*For correspondence:
daniel.knight@murdoch.edu.au (DRK);
thomas.riley@uwa.edu.au (TVR)

**Abstract** *Clostridioides difficile* infection (CDI) remains an urgent global One Health threat. The genetic heterogeneity seen across *C. difficile* underscores its wide ecological versatility and has driven the significant changes in CDI epidemiology seen in the last 20 years. We analysed an international collection of over 12,000 *C. difficile* genomes spanning the eight currently defined phylogenetic clades. Through whole-genome average nucleotide identity, and pangenomic and Bayesian analyses, we identified major taxonomic incoherence with clear species boundaries for each of the recently described cryptic clades CI–III. The emergence of these three novel genomospecies predates clades C1–5 by millions of years, rewriting the global population structure of *C. difficile* specifically and taxonomy of the *Peptostreptococcaceae* in general. These genomospecies all show unique and highly divergent toxin gene architecture, advancing our understanding of the evolution of *C. difficile* and close relatives. Beyond the taxonomic ramifications, this work may impact the diagnosis of CDI.

## Introduction

The bacterial species concept remains controversial, yet it serves as a critical framework for all aspects of modern microbiology (*Doolittle and Papke, 2006*). The prevailing species definition describes a genomically coherent group of strains sharing high similarity in many independent phenotypic and ecological properties (*Konstantinidis et al., 2006*). The era of whole-genome sequencing (WGS) has seen average nucleotide identity (ANI) replace DNA-DNA hybridisation as the next-generation standard for microbial taxonomy (*Wayne et al., 1987*; *Ciufo et al., 2018*). Endorsed by the National Center for Biotechnology Information (NCBI) (*Ciufo et al., 2018*), ANI provides a precise, objective, and scalable method for delineation of species, defined as monophyletic groups of strains with genomes that exhibit at least 96% ANI (*Jain et al., 2018*; *Richter and Rosselló-Móra, 2009*).

*Clostridioides (Clostridium) difficile* is an important gastrointestinal pathogen that places a significant growing burden on health-care systems in many regions of the world (*Guh et al., 2020*). In both its 2013 (*Centers for Disease Control and Prevention, 2013*) and 2019 (*Centers for Disease Control and Prevention, 2019*) reports on antimicrobial resistance (AMR), the US Centers for Disease Control and Prevention rated *C. difficile* infection (CDI) as an urgent health threat, the highest level. Community-associated CDI has become more frequent (*Guh et al., 2020*) and is linked to sources of *C. difficile* in animals and the environment (*Lim et al., 2020*). Thus, over the last two decades, CDI has emerged as an important One Health issue (*Lim et al., 2020*).

Based on multi-locus sequence type (MLST), there are eight recognised monophyletic groups or 'clades' of *C. difficile* (*Knight et al., 2015*). Strains within these clades show many unique clinical, microbiological, and ecological features (*Knight et al., 2015*). Critical to the pathogenesis of CDI is the expression of the large clostridial toxins, TcdA and TcdB, and, in some strains, binary toxin (CDT), encoded by two separate chromosomal loci, the PaLoc and CdtLoc, respectively (*Chandrasekaran and Lacy, 2017*). Clade 1 (C1) contains over 200 toxigenic and non-toxigenic sequence types (STs) including many of the most prevalent strains causing CDI worldwide, for example, ST2, ST8, and ST17 (*Knight et al., 2015*). Several highly virulent CDT-producing strains, including ST1 (PCR ribotype [RT] 027), a lineage associated with major hospital outbreaks in North America, Europe, and Latin America (*He et al., 2013*), are found in clade 2 (C2). Comparatively little is known about clade 3 (C3), although it contains ST5 (RT 023), a toxigenic CDT-producing strain with characteristics that may make laboratory detection difficult (*Shaw et al., 2020*). *C. difficile* ST37 (RT 017) is found in clade 4 (C4) and, despite the absence of a toxin A gene, is responsible for much of the endemic CDI burden in Asia (*Imwattana et al., 2019*). Clade 5 (C5) contains several CDT-producing strains including ST11 (RTs 078, 126, and others), which are highly prevalent in production animals worldwide (*Knight et al., 2019*). The remaining so-called 'cryptic' clades (C-I, C-II, and C-III), first described in 2012 (*Dingle et al., 2014*; *Didelot et al., 2012*), contain over 50 STs from clinical and environmental sources (*Dingle et al., 2014*; *Didelot et al., 2012*; *Janezic et al., 2016*; *Ramirez-Vargas and Rodriguez, 2020*; *Ramírez-Vargas et al., 2018*). The evolution of the cryptic clades is poorly understood. Clade C-I strains can cause CDI; however, due to atypical toxin gene architecture, they may not be detected, thus their prevalence may have been underestimated (*Ramírez-Vargas et al., 2018*).

There are over 600 STs currently described, and some STs may have access to a gene pool of more than 10,000 genes (*Knight et al., 2015*; *Knight et al., 2019*; *Knight et al., 2016*). Considering such enormous diversity, and recent contentious taxonomic revisions (*Lawson et al., 2016*; *Oren and Rupnik, 2018*), we hypothesise that *C. difficile* comprises a complex of distinct species divided along the major evolutionary clades. In this study, whole-genome ANI, and pangenomic and Bayesian analyses are used to explore an international collection of over 12,000 *C. difficile* genomes, to provide new insights into ancestry, genetic diversity, and evolution of pathogenicity in this enigmatic pathogen.

## Results

### An updated global population structure based on sequence typing of 12,000 genomes

We obtained and determined the ST and clade for a collection of 12,621 *C. difficile* genomes (taxid ID 1496, Illumina data) existing in the NCBI Sequence Read Archive (SRA) as of 1 January 2020. A total of 272 STs were identified spanning the eight currently described clades, indicating that the SRA contains genomes for almost 40% of known *C. difficile* STs worldwide (n = 659, PubMLST,

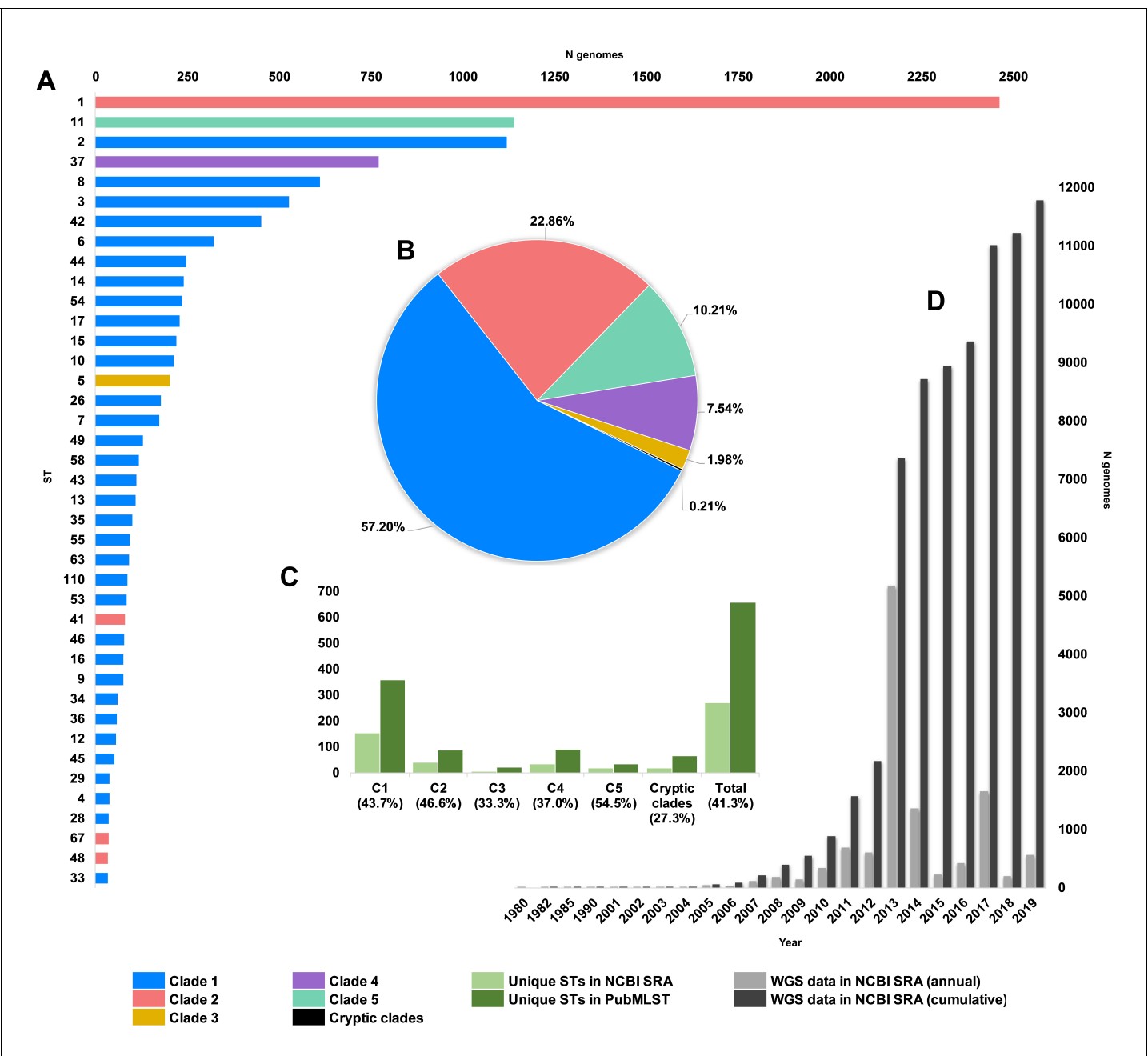

**Figure 1.** Composition of *C. difficile* genomes in the National Center for Biotechnology Information (NCBI) Sequence Read Archive (SRA). Snapshot obtained 1 January 2020; 12,304 strains (taxid ID 1496). (**A**) Top 40 most prevalent sequence types (STs) in the NCBI SRA coloured by clade. (**B**) The proportion of genomes in SRA by clade. (**C**) Number/proportion of STs per clade found in the SRA/present in the PubMLST database. (**D**) Annual and cumulative deposition of *C. difficile* genome data in SRA.

January 2020). C1 STs dominated the database in both prevalence and diversity (*Figure 1*) with 149 C1 STs comprising 57.2% of genomes, followed by C2 (35 STs, 22.9%), C5 (18 STs, 10.2%), C4 (34 STs, 7.5%), C3 (7 STs, 2.0%), and the cryptic clades C-I, C-II, and C-III (collectively 17 STs, 0.2%). The five most prevalent STs represented were ST1 (20.9% of genomes), ST11 (9.8%), ST2 (9.5%), ST37 (6.5%), and ST8 (5.2%), all prominent lineages associated with CDI worldwide (*Knight et al., 2015*).

*Figure 2* shows an updated global *C. difficile* population structure based on the 659 STs; 27 novel STs were found (an increase of 4%) and some corrections to assignments within C1 and C2 were made, including assigning ST122 (*Knetsch et al., 2012*) to C1. Based on PubMLST data and boot-straps values of 1.0 in all monophyletic nodes of the cryptic clades (*Figure 2*), we could confidently assign 25, 9, and 10 STs to cryptic clades I, II, and III, respectively. There remained 26 STs spread across the phylogeny that did not fit within a specific clade (defined as outliers). The full MLST data

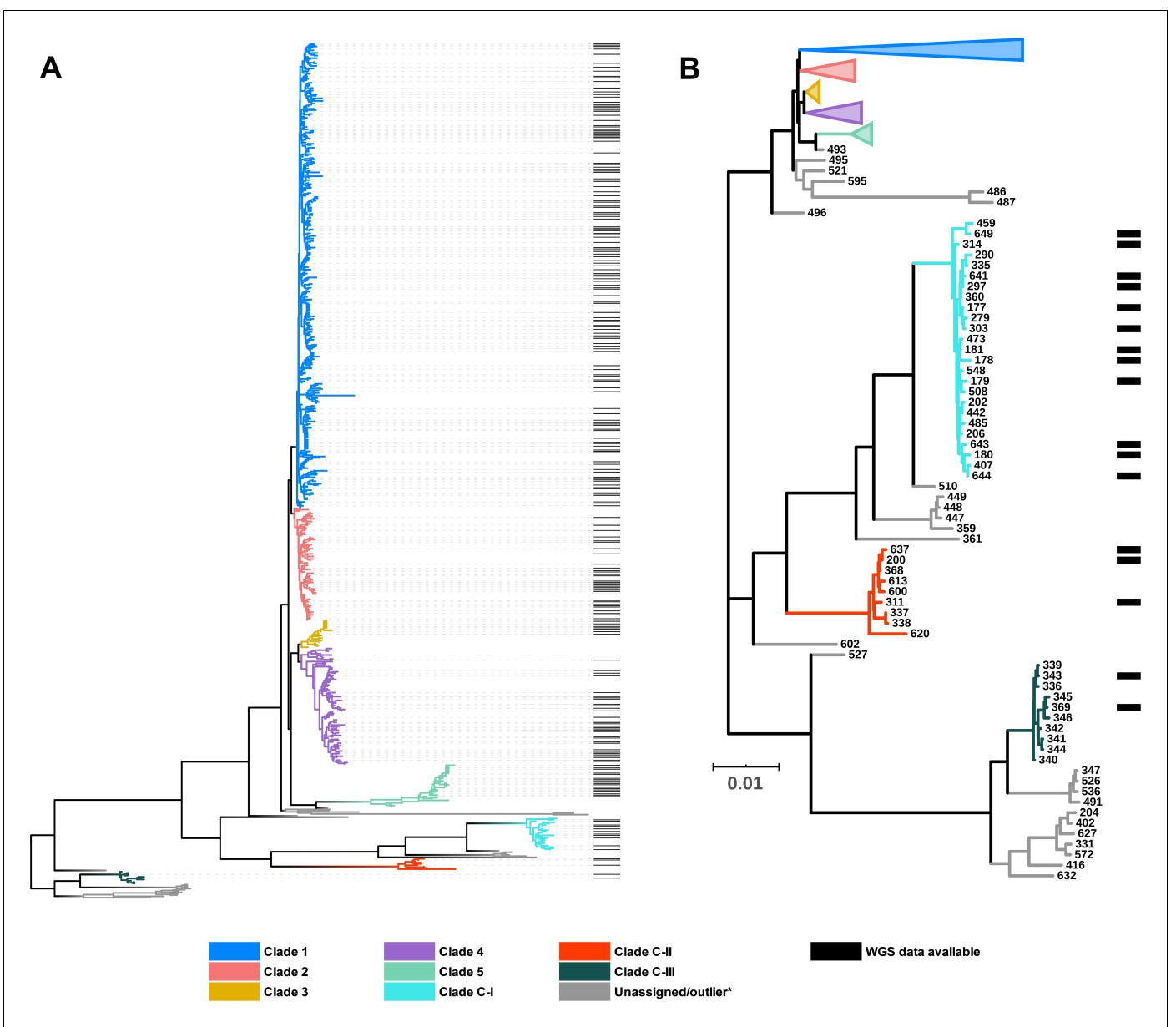

**Figure 2.** *C. difficile* population structure. (A) Neighbor joining phylogeny of 659 aligned, concatenated, multilocus sequence-type (MLST) allele combinations coloured by current PubMLST clade assignment. Black bars indicate whole-genome sequencing (WGS) available for average nucleotide identity (ANI) analysis (n = 260). (B) A subset of the tree showing cryptic clades C-I, C-II, and C-III. Again, black bars indicate WGS available for ANI analysis (n = 17).

and tree file for *Figure 2* are available as *Supplementary files 1a–d* and *2* at http://doi.org/10.6084/m9.figshare.12471461. Representative genomes of each ST present in the SRA were chosen based on metadata, read depth, and assembly quality. This resulted in a final dataset of 260 STs (C1, n = 149; C2, n = 35; C3, n = 7; C4, n = 34; C5, n = 18; C-I, n = 12; C-II, n = 3; C-III, n = 2) used for all subsequent bioinformatics analyses. The list of representative genomes is available in *Supplementary file 1b*.

## Whole-genome ANI analysis reveals clear species boundaries

Whole-genome ANI analyses were used to investigate genetic discontinuity across the *C. difficile* species (*Figure 3* and *Supplementary file 1f*). Whole-genome ANI values were determined for the final set of 260 STs using three independent ANI algorithms (FastANI, ANIm, and ANIb; see Materials and methods). All 225 STs belonging to clades C1–4 clustered within an ANI range of 97.1–99.8% (median FastANI values of 99.2, 98.7, 97.9, and 97.8%, respectively; *Figure 3A–C*).

These ANI values are above the 96% species demarcation threshold used by the NCBI (*Ciufo et al., 2018*) and indicate that strains from these clades belong to the same species. ANI values for all 18 STs belonging to C5 clustered on the borderline of the species demarcation threshold (FastANI range 95.9–96.2%, median 96.1%). ANI values for all three cryptic clades fell well below the species threshold; C-I (FastANI range 90.9–91.1%, median 91.0%), C-II (FastANI range 93.6–93.9%, median 93.7%), and C-III (FastANI range 89.1–89.1%, median 89.1%). All results were corroborated across the three independent ANI algorithms (*Figure 3A–C*). *C. difficile* strain ATCC 9689 (ST3, C1) was defined by Lawson et al. as the type strain for the species (*Lawson et al., 2016*) and used as a reference in all the above analyses. To better understand the diversity among the divergent clades themselves, FastANI analyses were repeated using STs 11, 181, 200, and 369 as reference archetypes of clades C5, C-I, C-II, and C-III, respectively. This approach confirmed that C5 and the three cryptic clades were as distinct from each other as they were collectively from C1–4 (*Figure 3D–G*).

## Taxonomic placement of cryptic clades predates *C. difficile* emergence by millions of years

Previous studies using BEAST have estimated the common ancestor of C1–5 existed between 1 to 85 or 12 to 14 million years ago (mya) (*He et al., 2010*; *Kumar et al., 2019*). Here, we used an alternative Bayesian approach, BactDating, to estimate the age of all eight *C. difficile* clades currently described. The last common ancestor for *C. difficile* clades C1–5 was estimated to have existed between 1.11 and 6.71 mya. In contrast, all three cryptic clades were estimated to have emerged millions of years prior to the common ancestor of C1–5 (*Figure 4*). Independent analysis with BEAST, using a smaller core gene dataset (see Materials and methods), provided temporal estimates of clade emergence that were of the same order of magnitude and, importantly, supported the same branching order for all clades (*Figure 4*).

Next, to identify their true taxonomic placement, ANI was determined for ST181 (C-I), ST200 (C-II), and ST369 (C-III) against two reference datasets. The first dataset comprised 25 species belonging to the *Peptostreptococcaceae* as defined by *Lawson et al., 2016* in their 2016 reclassification of *Clostridium difficile* to *Clostridioides difficile*. The second dataset comprised 5895 complete genomes across 21 phyla from the NCBI RefSeq database (accessed 14 January 2020), including 1366 genomes belonging to *Firmicutes*, 92 genomes belonging to 15 genera within the *Clostridiales*, and 18 *Clostridium* and 2 *Clostridioides* species. The nearest ANI matches to species within the *Peptostreptococcaceae* dataset were *C. difficile* (range 89.3–93.5% ANI), *Asaccharospora irregularis* (78.9–79.0% ANI), and *Romboutsia lituseburensis* (78.4–78.7% ANI). Notably, *Clostridioides mangenotii*, the only other known member of *Clostridioides*, shared only 77.2–77.8% ANI with the cryptic clade genomes (*Table 1*).

Similarly, the nearest ANI matches to species within the RefSeq dataset were several *C. difficile* strains (range C-I: 90.9–91.1%; C-II: 93.4–93.6%; and C-III: 89.2–89.4%) and *Paeniclostridium sordellii* (77.7–77.9%). A low ANI (range ≤70–75%) was observed between the cryptic clade genomes and 20 members of the *Clostridium* including *Clostridium tetani*, *Clostridium botulinum*, *Clostridium perfringens*, and *Clostridium butyricum*, the type strain of the *Clostridium* genus *senso stricto*. An updated ANI-based taxonomy for the *Peptostreptococcaceae* is shown in *Figure 5A*. The phylogeny places C-I, C-II, and C-III between *C. mangenotii* and *C. difficile* C1–5, suggesting that they should be

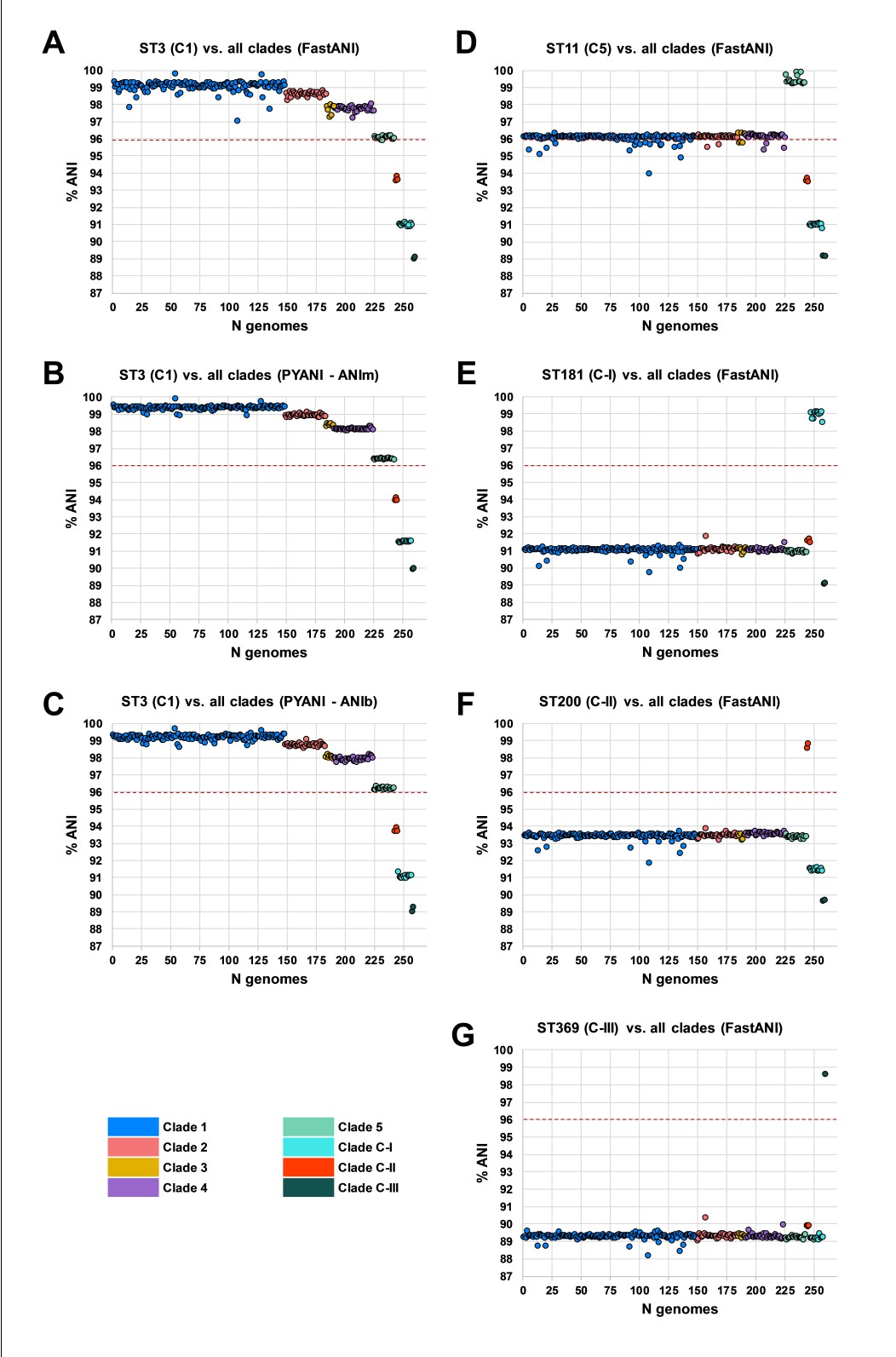

**Figure 3.** Species-wide average nucleotide identity (ANI) analysis. Panels (**A–C**) show ANI plots for sequence type (ST)3 (C1) vs. all clades (260 STs) using FastANI, ANIm, and ANIb algorithms, respectively. Panels (**D–G**) show ANI plots for ST11 (C5), ST181 (C-I), ST200 (C-II), and ST369 (C-III) vs. all clades (260 STs), respectively. National Center for Biotechnology Information species demarcation of 96% indicated by red dashed line (**Ciufo et al., 2018**).

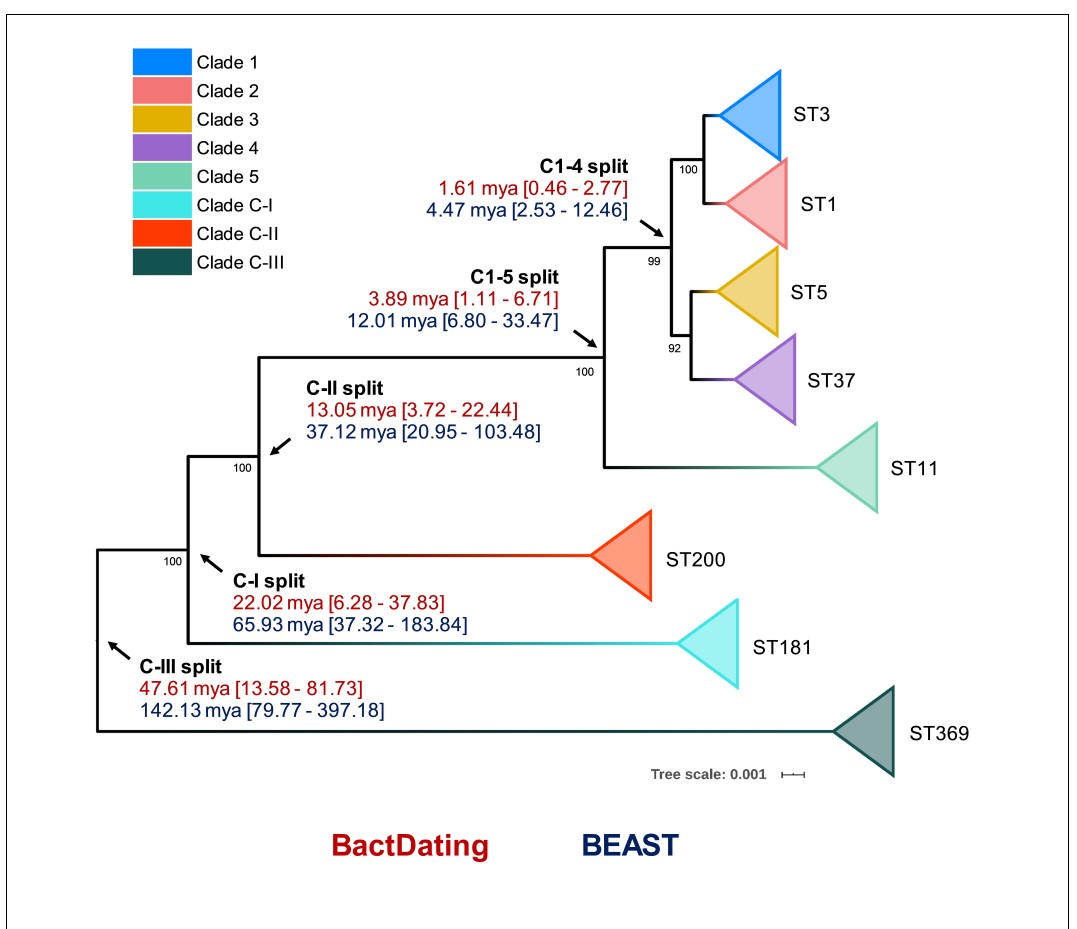

**Figure 4.** Bayesian analysis of species and clade divergence. BactDating and BEAST estimates of the age of major *C. difficile* clades. Node dating ranges for both Bayesian approaches are transposed onto an maximum-likelihood phylogeny built from concatenated multi-locus sequence type (MLST) alleles of a dozen sequence types (STs) from each clade. Archetypal STs in each evolutionary clade are indicated. The tree is midpoint rooted, and bootstrap values are shown (all bootstrapping values of the cryptic clade branches are 100%). Scale bar indicates the number of substitutions per site. BactDating estimates the median time of the most recent common ancestor of C1–5 at 3.89 million years ago (mya) (95% credible interval [CI], 1.11–6.71 mya). Of the cryptic clades, C-II shared the most recent common ancestor with C1–5 (13.05 mya, 95% CI 3.72–22.44 mya), followed by C-I (22.02 mya, 95% CI 6.28–37.83 mya) and C-III (47.61 mya, 95% CI 13.58–81.73 mya). Comparative temporal estimates from BEAST show the same order of magnitude and support the same branching order (clades C1–5 [12.01 mya, 95% CI 6.80–33.47 mya]; C-II [37.12 mya, 95% CI 20.95–103.48 mya]; C-I [65.93 mya, 95% CI 37.32–183.84 mya]; C-III [142.13 mya, 95% CI 79.77–397.18 mya]).

assigned to the *Clostridioides* genus, distinct from both *C. mangenotii* and *C. difficile*. Comparative analysis of ANI and 16S rRNA values for the eight *C. difficile* clades and *C. mangenotii* shows significant incongruence between the data generated by the two approaches (*Figure 5B*). The range of 16S rRNA % similarity between *C. difficile* C1–4, cryptic clades I–III, and *C. mangenotii* was narrower (range 94.5–100) compared to the range of ANI values (range 77.8–98.7). Curiously, *C. mangenotii* and *C. difficile* shared 94.5–94.7% similarity in 16S rRNA sequence identity, yet only 77.8–78.2% ANI, indicating that they should not even be considered within the same genus, as proposed by *Lawson et al., 2016*.

We also extended our approach to five other medically important clostridia available on the NCBI database; *C. botulinum* (n = 783), *C. perfringens* (n = 358), *Clostridium sporogenes* (n = 100), *C. tetani* (n = 32), and *P. sordellii* (formerly *Clostridium sordellii*, n = 46). We found that three out of the five species (*C. perfringens*, *C. sporogenes*, and *C. botulinum*) showed evidence of taxonomic discontinuity similar to that observed for *C. difficile* (e.g., a proportion of strains with pairwise ANI

Table 1. Whole-genome ANI analysis of cryptic clades vs. 25 *Peptostreptococcaceae* species from *Lawson et al., 2016*.

| Species | NCBI accession | ANI % | | |
| --- | --- | --- | --- | --- |
| | | ST181 (C-I) | ST200 (C-II) | ST369 (C-III) |
| *Clostridioides difficile* (ST3) | AQWV00000000.1 | 91.11 | 93.54 | 89.30 |
| *Asaccharospora irregularis* | NZ_FQWX00000000 | 78.94 | 78.87 | 78.91 |
| *Romboutsia lituseburensis* | NZ_FNGW00000000.1 | 78.51 | 78.36 | 78.66 |
| *Romboutsia ilealis* | LN555523.1 | 78.45 | 78.54 | 78.44 |
| *Paraclostridium benzoelyticum* | NZ_LBBT00000000.1 | 77.92 | 77.71 | 78.14 |
| *Paraclostridium bifermentans* | NZ_AVNC00000000.1 | 77.89 | 77.89 | 78.06 |
| *Clostridioides mangenotii* | GCA_000687955.1 | 77.82 | 77.84 | 78.15 |
| *Paeniclostridium sordellii* | NZ_APWR00000000.1 | 77.73 | 77.59 | 77.86 |
| *Clostridium hiranonis* | NZ_ABWP01000000 | 77.52 | 77.42 | 77.59 |
| *Terrisporobacter glycolicus* | NZ_AUUB00000000.1 | 77.47 | 77.53 | 77.53 |
| *Intestinibacter bartlettii* | NZ_ABEZ00000000.2 | 77.29 | 77.52 | 77.48 |
| *Clostridium paradoxum* | NZ_LSFY00000000.1 | 76.60 | 76.65 | 76.93 |
| *Clostridium thermoalcaliphilum* | NZ_MZGW00000000.1 | 76.49 | 76.61 | 76.85 |
| *Tepidibacter formicigenes* | NZ_FRAE00000000.1 | 76.41 | 76.47 | 76.38 |
| *Tepidibacter mesophilus* | NZ_BDQY00000000.1 | 76.38 | 76.44 | 76.22 |
| *Tepidibacter thalassicus* | NZ_FQXH00000000.1 | 76.34 | 76.31 | 76.46 |
| *Peptostreptococcus russellii* | NZ_JYGE00000000.1 | 76.30 | 76.08 | 76.38 |
| *Clostridium formicaceticum* | NZ_CP020559.1 | 75.18 | 75.26 | 75.62 |
| *Clostridium caminithermale* | FRAG00000000 | 74.97 | 75.07 | 75.03 |
| *Clostridium aceticum* | NZ_JYHU00000000.1 | ≤70.00 | ≤70.00 | ≤70.00 |
| *Clostridium litorale* | FSRH01000000 | ≤70.00 | ≤70.00 | ≤70.00 |
| *Eubacterium acidaminophilum* | NZ_CP007452.1 | ≤70.00 | ≤70.00 | ≤70.00 |
| *Filifactor alocis* | NC_016630.1 | ≤70.00 | ≤70.00 | ≤70.00 |
| *Peptostreptococcus anaerobius* | ARMA01000000 | ≤70.00 | ≤70.00 | ≤70.00 |
| *Peptostreptococcus stomatis* | NZ_ADGQ00000000.1 | ≤70.00 | ≤70.00 | ≤70.00 |

NCBI: National Center for Biotechnology Information; ANI: average nucleotide identity: ST: sequence type.

below the 96% demarcation threshold). This was most notable for *C. sporogenes* and *C. botulinum*, where there were many sequenced strains with a pairwise ANI below 90% (8% and 31% of genomes, respectively, *Supplementary file 1i*).

## Evolutionary and ecological insights from the *C. difficile* species pangenome

Next, we sought to quantify the *C. difficile* species pangenome and identify genetic loci that are significantly associated with the taxonomically divergent clades. With Panaroo, the *C. difficile* species pangenome comprised 17,470 genes, encompassing an accessory genome of 15,238 genes and a core genome of 2232 genes, just 12.8% of the total gene repertoire (*Figure 6*). The size of the pangenome reduced by 2082 genes with the exclusion of clades CI-III, and a further 519 genes with the exclusion of C5. Compared to Panaroo, Roary overestimated the size of the pangenome (32,802 genes, 87.7% overestimation), resulting in markedly different estimates of the percentage core genome, 3.9% and 12.8%, respectively ($\chi^2$ = 1395.3, df = 1, p<0.00001). The overestimation of pangenome was less pronounced when the identity threshold was decreased to 90% (42.0% overestimation) and the paralogs were merged (28.7% overestimation). Panaroo can account for errors

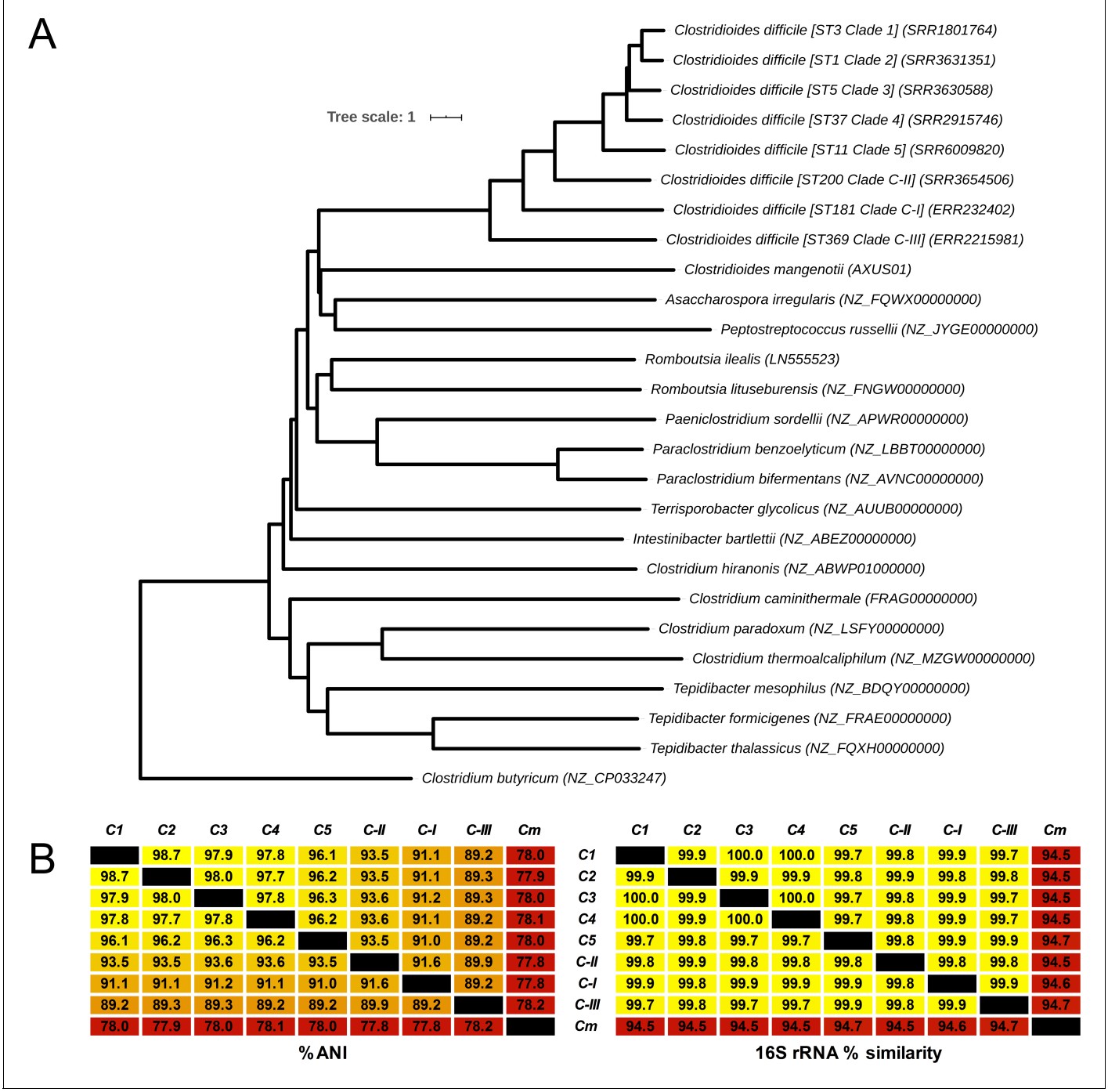

**Figure 5.** Revised taxonomy for the *Peptostreptococcaceae*. (**A**) Average nucleotide identity (ANI)-based minimum evolution tree showing evolutionary relationship between 8 *C. difficile* 'clades' along with 17 members of the *Peptostreptococcaceae* (from *Lawson et al., 2016*) as well as *Clostridium butyricum* as the outgroup and type strain of the *Clostridium* genus *senso stricto*. To convert the ANI into a distance, its complement to 1 was taken. (**B**) Matrices showing pairwise ANI and 16S rRNA values for the eight *C. difficile* clades and *C. mangenotii* (*Cm*), the only other known member of *Clostridioides*.

introduced during assembly and annotation, thus polishing the 260 Prokka-annotated genomes with Panaroo resulting in a significant reduction in gene content per genome (median 2.48%; 92 genes, range 1.24–12.40%; 82–107 genes, p<0.00001). The *C. difficile* species pangenome was determined to be open (*Tettelin et al., 2005*; *Figure 6*).

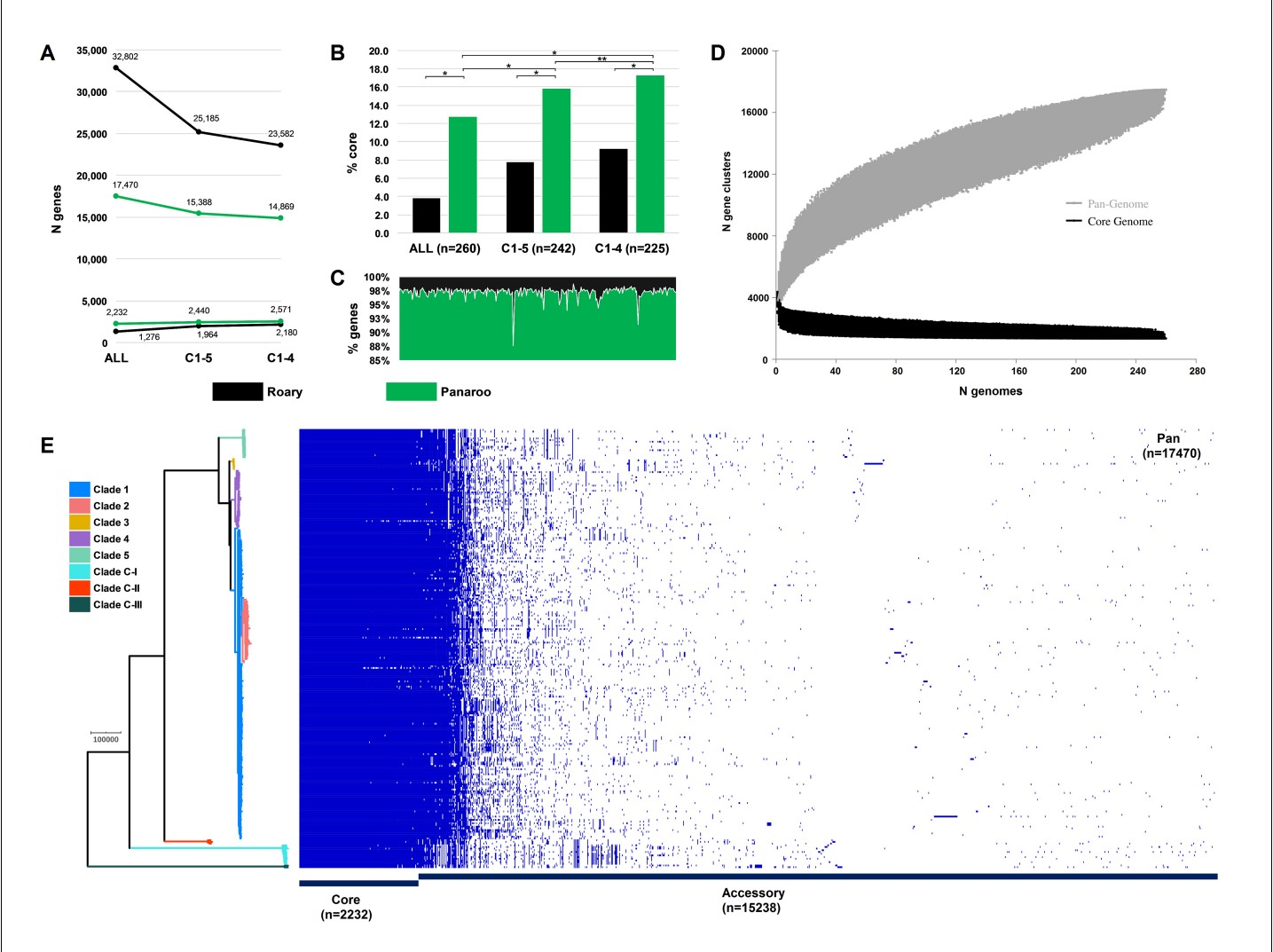

**Figure 6.** *Clostridioides difficile* species pangenome. (**A**) Pan and core genome estimates for all 260 sequence types (STs), clades C1–4 (n = 242 STs) and clades C1–5 (n = 225 STs). (**B**) The difference in % core genome and pangenome sizes with Panaroo and Roary algorithms. * indicates χ² p<0.00001 and ** indicates χ² p=0.0008. (**C**) The proportion of retained genes per genome after polishing Prokka-annotated genomes with Panaroo. (**D**) The total number of genes in the pan (grey) and core (black) genomes is plotted as a function of the number of genomes sequentially added (n = 260). Following the definition of *Tettelin et al., 2005.*, the *C. difficile* species pangenome showed characteristics of an 'open' pangenome. First, the pangenome increased in size exponentially with sampling of new genomes. At n = 260, the pangenome exceeded more than double the average number of genes found in a single *C. difficile* genome (~3700) and the curve was yet to reach a plateau or exponentially decay, indicating more sequenced strains are needed to capture the complete species gene repertoire. Second, the number of new 'strain-specific' genes did not converge to zero upon sequencing of additional strains, at n = 260, an average of 27 new genes were contributed to the gene pool. Finally, according to Heap's law, α values of ≤1 are representative of open pangenome. Rarefaction analysis of our pangenome curve using a power-law regression model based on Heap's law (*Tettelin et al., 2005*) showed the pangenome was predicted to be open (*Bpan* [≈ α (*Tettelin et al., 2005*) = 0.47], curve fit, $r^2$ = 0.999). (**E**) Presence-absence variation (PAV) matrix for 260 *C. difficile* genomes is shown alongside a maximum-likelihood phylogeny built from a recombination-adjusted alignment of core genes from Panaroo (2232 genes, 2,606,142 sites).

Pangenome-Wide Association Study (Pan-GWAS) analysis with Scoary revealed 142 genes with significant clade specificity. Based on KEGG orthology, these genes were classified into four functional categories: environmental information processing, genetic information processing, metabolism, and signalling and cellular processes. We identified several uniquely present, absent, or organised gene clusters associated with ethanolamine catabolism (C-III), heavy metal uptake (C-III), polyamine biosynthesis (C-III), fructosamine utilisation (C-I, C-III), zinc transport (C-II, C5), and folate metabolism (C-I, C5). A summary of the composition and function of these major lineage-specific

**Table 2.** Major clade-specific gene clusters identified by Pangenome-Wide Association Study (pan-GWAS).

| Protein | Gene | Clade specificity | Functional insights |
|---|---|---|---|
| Ethanolamine kinase | ETNK, EKI | Unique to C-III and is in addition to the highly conserved eut cluster found in all lineages. Has a unique composition and includes six additional genes that are not present in the traditional CD630 eut operon or any other non-C-III strains. | An alternative process for the breakdown of ethanolamine and its utilisation as a source of reduced nitrogen and carbon. |
| Agmatinase | speB | | |
| 1-propanol dehydrogenase | pduQ | | |
| Ethanolamine utilisation protein EutS | eutS | | |
| Ethanolamine utilisation protein EutP | eutP | | |
| Ethanolamine ammonia-lyase large subunit | eutB | | |
| Ethanolamine ammonia-lyase small subunit | eutC | | |
| Ethanolamine utilisation protein EutL | eutL | | |
| Ethanolamine utilisation protein EutM | eutM | | |
| Acetaldehyde dehydrogenase | E1.2.1.10 | | |
| Putative phosphotransacetylase | K15024 | | |
| Ethanolamine utilisation protein EutN | eutN | | |
| Ethanolamine utilisation protein EutQ | eutQ | | |
| TfoX/Sxy family protein | - | | |
| Iron complex transport system permease protein | ABC.FEV.P | Unique to C-III. | Multicomponent transport system with specificity for chelating heavy metal ions. |
| Iron complex transport system ATP-binding protein | ABC.FEV.A | | |
| Iron complex transport system substrate-binding protein | ABC.FEV.S | | |
| Hydrogenase nickel incorporation protein HypB | hypB | | |
| Putative ABC transport system ATP-binding protein | yxdL | | |
| Class I SAM-dependent methyltransferase | - | | |
| Peptide/nickel transport system substrate-binding protein | ABC.PE.S | | |
| Peptide/nickel transport system permease protein | ABC.PE.P | | |
| Peptide/nickel transport system permease protein | ABC.PE.P1 | | |
| Peptide/nickel transport system ATP-binding protein | ddpD | | |
| Oligopeptide transport system ATP-binding protein | oppF | | |
| Class I SAM-dependent methyltransferase | - | | |
| Heterodisulfide reductase subunit D (EC:1.8.98.1) | hdrD | Unique to C-III and is in addition to the highly conserved spermidine uptake cluster found in all other lineages. | Alternative spermidine uptake processes that may play a role in stress response to nutrient limitation. The additional cluster has homologs in *Romboutsia*, *Paraclostridium*, and *Paeniclostridium* spp. |
| CDP-L-myo-inositol myo-inositolphosphotransferase | dipps | | |
| Spermidine/putrescine transport system substrate-binding protein | ABC.SP.S | | |
| Spermidine/putrescine transport system permease protein | ABC.SP.P1 | | |
| Spermidine/putrescine transport system permease protein | ABC.SP.P | | |
| Spermidine/putrescine transport system ATP-binding protein | potA | | |
| Sigma-54-dependent transcriptional regulator | gfrR | Present in all lineages except C-I. Cluster found in a different genomic position in C-III. | Mannose-type PTS system essential for utilisation of fructosamines such as fructoselysine and glucoselysine, abundant components of rotting fruit and vegetable matter. |
| Fructoselysine/glucoselysine PTS system EIIB component | gfrB | | |
| Mannose PTS system EIIA component | manXa | | |
| Fructoselysine/glucoselysine PTS system EIIC component | gfrC | | |
| Fructoselysine/glucoselysine PTS system EIID component | gfrD | | |
| SIS domain-containing protein | - | | |
| Fur family transcriptional regulator, ferric uptake regulator | furB | Unique to C-II and C5. | Associated with EDTA resistance in *E. coli*, helping the bacteria survive in Zn-depleted environment. |
| Zinc transport system substrate-binding protein | znuA | | |
| Fe-S-binding protein | yeiR | | |
| Rrf2 family transcriptional regulator | - | | |
| Putative signalling protein | - | Unique to C-I and C5 STs 163, 280, and 386 | In *E. coli*, AbgAB proteins enable uptake and cleavage of the folate catabolite *p*-aminobenzoyl-glutamate, allowing the bacterium to survive on exogenous sources of folic acid. |
| Aminobenzoyl-glutamate utilisation protein B | abgB | | |
| MarR family transcriptional regulator | - | | |

gene clusters is given in *Table 2*, and a comparative analysis of their respective genetic architecture can be found in *Supplementary file 1l*.

## Cryptic clades CI-III possessed highly divergent toxin gene architecture

Overall, 68.8% (179/260) of STs harboured *tcdA* (toxin A) and/or *tcdB* (toxin B), the major virulence factors in *C. difficile*, while 67 STs (25.8%) harboured *cdtA*/*cdtB* (binary toxin). The most common genotype was A$^+$B$^+$CDT$^-$ (113/187; 60.4%), followed by A$^+$B$^+$CDT$^+$ (49/187; 26.2%), A$^-$B$^+$CDT$^+$ (10/187; 5.3%), A$^-$B$^-$CDT$^+$ (8/187; 4.3%), and A$^-$B$^+$CDT$^-$ (7/187; 3.7%). Toxin gene content varied across clades (C1, 116/149, 77.9%; C2, 35/35, 100.0%; C3, 7/7, 100.0%; C4, 6/34, 17.6%; C5, 18/18, 100.0%; C-I, 2/12, 16.7%; C-II, 1/3, 33.3%; C-III, 2/2, 100.0%) (*Figure 7*).

Critically, at least one ST in each of clades C-I, C-II, and C-III harboured divergent *tcdB* (89–94% identity to *tcdB*$_{R20291}$) and/or *cdtAB* alleles (60–71% identity to *cdtA*$_{R20291}$, 74–81% identity to *cdtB*$_{R20291}$). These genes were located on atypical and novel PaLoc and CdtLoc structures flanked by mediators of lateral gene transfer (*Figure 7*). STs 359, 360, 361, and 649 (C-I), 637 (C-II), and 369 (C-III) harboured 'monotoxin' PaLocs characterised by the presence of syntenic *tcdR*, *tcdB,* and *tcdE*, and complete absence of *tcdA* and *tcdC*. In STs 360 and 361 (C-I), and 637 (C-II), a gene encoding an endolysin with predicted N-acetylmuramoyl-L-alanine amidase activity (*cwlH*) was found adjacent to the phage-derived holin gene *tcdE*.

Remarkably, a full CdtLoc was found upstream of the PaLoc in ST369 (C-III). This CdtLoc was unusual, characterised by the presence of *cdtB*, two copies of *cdtA*, two copies of *cdtR* and *xerC* encoding a site-specific tyrosine recombinase (*Figure 7*). Both ST644 (C-I) and ST343 (C-III) were CdtLoc-positive but PaLoc-negative (A$^-$B$^-$CDT$^+$). In ST649 (C-I), *cdtR* was completely absent, and in ST343 (C-III), the entire CdtLoc was contained within the genome of a 56 kbp temperate bacteriophage termed ΦSemix9P1 (*Riedel et al., 2017*). Toxin regulators TcdR and CdtR are highly conserved across clades C1–5 (*Ramírez-Vargas et al., 2018*). In contrast, the CdtR of STs 644 (C-I), 343 (C-III), and 369 (C-III) shared only 46–54% amino acid identity (AAI) with CdtR of strain R20291 from clade 2 and ~40% AAI to each other. Similarly, the TcdR of ST 369 shared only 82.1% AAI compared to R20291 (*Supplementary file 1m*).

Compared to TcdB of R20291 (TcdB$_{R20291}$), the shared AAI for TcdB$_{ST649\_C-I}$, TcdB$_{ST637\_C-II}$, and TcdB$_{ST369\_C-III}$ were 94.0, 90.5, and 89.4%, respectively. This sequence heterogeneity was confirmed through the detection of five distinct *Hinc*II/*Acc*I digestion profiles of *tcdB* B1 fragments possibly reflecting novel toxinotypes (*Supplementary file 1n*). TcdB phylogenies identified clade C2 as the most recent common ancestor for TcdB$_{ST649\_C-I}$ (*Figure 7*). Phylogenetic subtyping analysis of the TcdB receptor-binding domain (RBD) showed the respective sequences in C-I, C-II, and C-III clustered with *tcdB* alleles belonging to virulent C2 strains (*Supplementary file 1o*). Notably, the TcdB-RBD of ST649 (C-I) shared an AAI of 93.5% with TcdB-RBD allele type 8 belonging to hypervirulent STs 1 (RT027) (*He et al., 2013*) and 231 (RT251) (*Hong et al., 2019*). Similarly, the closest match to TcdB-RBDs of ST637 (C-II) and ST369 (C-III) was allele type 10 (ST41, RT244, C2) (*Eyre et al., 2015*).

## Discussion

Through phylogenomic analysis of the largest and most diverse collection of *C. difficile* genomes to date, we identified major incoherence in *C. difficile* taxonomy, provide the first WGS-based phylogeny for the *Peptostreptococcaceae,* and provide new insight into intra-species diversity and evolution of pathogenicity in this major One Health pathogen.

Our analysis found high nucleotide identity (ANI >97%) between *C. difficile* clades C1–4, indicating that strains from these four clades (comprising 560 known STs) belong to the same species. On the other hand, ANI between C5 and C1–4 is on the borderline of the accepted species threshold (95.9–96.2%). This degree of speciation likely reflects the unique ecology of C5 – a lineage comprising 33 known STs, which is well established in non-human animal reservoirs worldwide and associated with CDI in the community setting (*Knight and Riley, 2019*). Conversely, we identified major taxonomic incoherence among the three cryptic clades and C1–5, evident by ANI values (compared to ST3, C1) far below the species threshold (~91%, C-I; ~94%, C-II; and ~89%, C-III). Similar ANI value

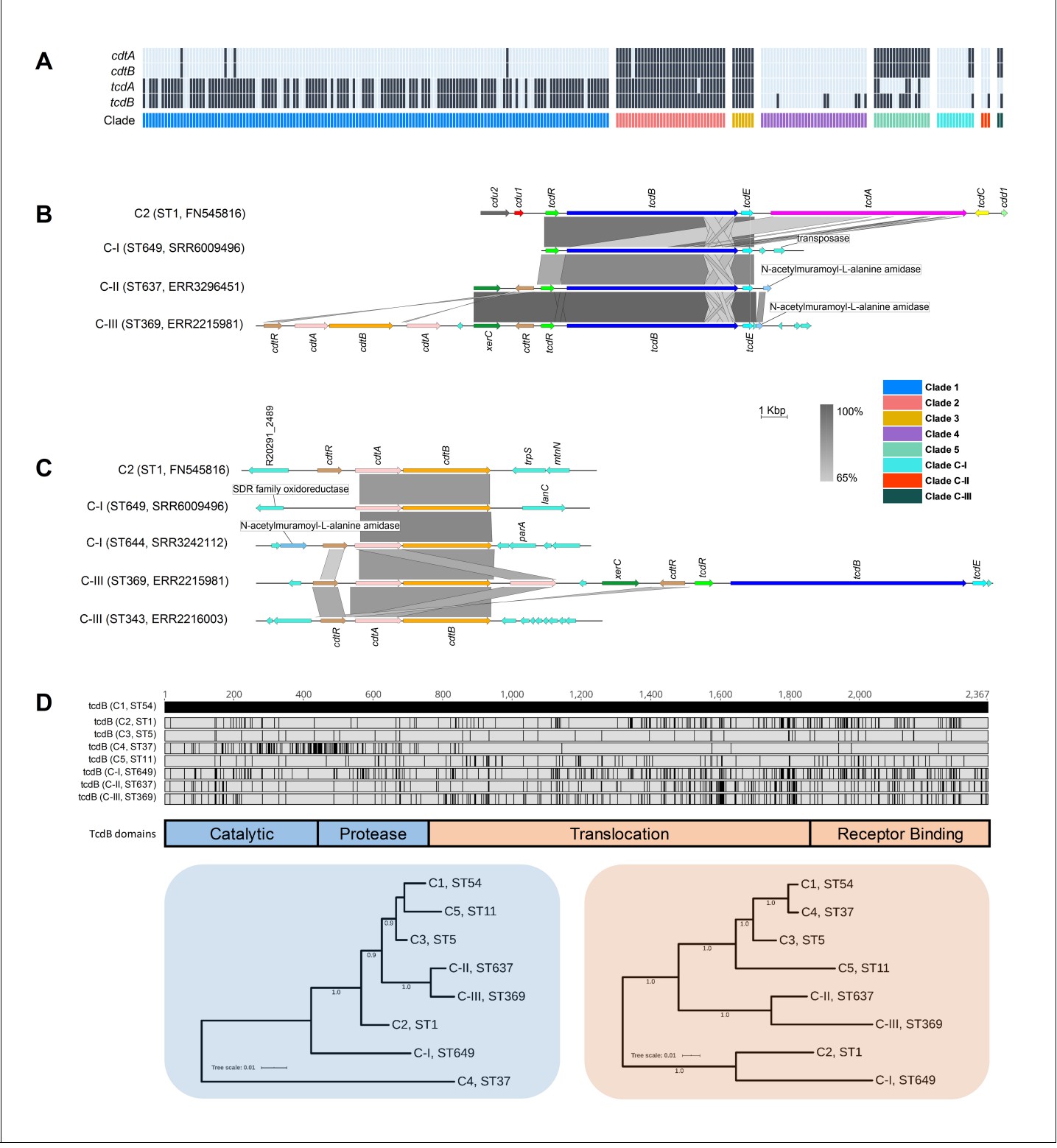

**Figure 7.** Toxin gene analysis. (**A**) Distribution of toxin genes across *C. difficile* clades (n = 260 sequence types [STs]). Presence is indicated by black bars and absence by light blue bars. (**B**) Comparison of PaLoc architecture in the chromosome of strain R20291 (C2, ST1) and cognate chromosomal regions in genomes of cryptic STs 649 (C-I), 637 (C-II), and 369 (C-III). All three cryptic STs show atypical 'monotoxin' PaLoc structures, with the presence of syntenic *tcdR*, *tcdB*, and *tcdE*, and the absence of *tcdA*, *tcdC*, *cdd1*, and *cdd2*. ST369 genome ERR2215981 shows colocalisation of the PaLoc and CdtLoc, see below. (**C**) Comparison of CdtLoc architecture in the chromosome of strain R20291 (C2, ST1) and cognate chromosomal regions in genomes of cryptic STs 649/644 (C-I) and 343/369 (C-III). Several atypical CdtLoc features are observed; *cdtR* is absent in ST649, and an additional copy

*Figure 7 continued on next page*

*Figure 7 continued*

of *cdtA* is present in ST369, the latter comprising part of a CdtLoc colocated with the PaLoc. (D) Amino acid differences in TcdB among cryptic STs 649, 637, and 369 and reference strains from clades C1–5. Variations are shown as black lines relative to CD630 (C1, ST54). Phylogenies constructed from the catalytic and protease domains (in blue) and translocation and receptor-binding domains (in orange) of TcdB for the same eight STs included in (D). Scale bar shows the number of amino acid substitutions per site. Trees are midpoint rooted and supported by 500 bootstrap replicates.

differences were seen between the cryptic clades themselves, indicating that they are as divergent from each other as they are individually from C1–5. This extraordinary level of discontinuity is substantiated by our core genome and Bayesian analyses. Our study estimated the most recent common ancestor of *C. difficile* clades C1–4 and C1–5 existed between 0.46 to 2.77 mya and between 1.11 to 6.71 mya, respectively, whereas the common ancestors of clades C-I, C-II, and C-III were estimated to have existed at least 1.5–75 million years before the common ancestor of C1–5. For context, divergence dates for other notable pathogens range from 10 million years (Ma) (*Campylobacter coli* and *C. jejuni*) (*Sheppard and Maiden, 2015*), 47 Ma (*Burkholderia pseudomallei* and *B. thailandensis*) (*Yu et al., 2006*), and 120 Ma (*Escherichia coli* and *Salmonella enterica*) (*Ochman et al., 1999*). Corresponding whole-genome ANI values for these species are 86, 94, and 82%, respectively (*Supplementary file 1j*).

Although BEAST provided wider confidence intervals (and therefore less certainty compared to BactDating), it estimates the time of divergence for all clades within the same order of magnitude and, importantly, provides robust support for the same branching order of clades with clade C-III the most ancestral of lineages, followed by the emergence of C-I, C-II, and C5. After this point, there appears to have been rapid population expansion into the four closely related clades described today, which include many of the most prevalent strains causing healthcare-associated CDI worldwide (*Knight et al., 2015*). We acknowledge that the dating of ancient taxa is often imprecise and that using a strict clock model for such a diverse set of taxa leads to considerable uncertainty in divergence estimates. However, we tried to mitigate this as much as possible by using two independent tools and evaluated multiple molecular clock estimates (covering almost an order of magnitude), ultimately using the same fixed clock model as *Kumar et al., 2019* ($2.5 \times 10^{-9}$–$10.5 \times 10^{-8}$). The branching order of the clades is robust, supported by comprehensive and independent comparative genomic and phylogenomic analyses. Notwithstanding this finding, if variations in the molecular clock happen over time and across lineages, which is likely the case for such a genetically diverse spore-forming pathogen, then the true age ranges for *C. difficile* clade emergence are likely far greater (and therefore less certain) than we report here.

Comparative ANI analysis of the cryptic clades with >5000 reference genomes across 21 phyla failed to provide a better match than *C. difficile* (89–94% ANI). Similarly, our revised ANI-based taxonomy of the *Peptostreptococcaceae* placed clades C-I, C-II, and C-III between *C. difficile* and *C. mangenotii*. Our analyses of the *Clostridioides* spp. highlights the major discordance between WGS data and 16S rRNA data, which has historically been used to classify bacterial species. In 2016, *Lawson et al., 2016* used 16S rRNA data to categorise *C. difficile* and *C. mangenotii* as the sole members of the *Clostridioides*. These species have 94.7% similarity in 16S rRNA sequence identity, yet our findings indicate that *C. mangenotii* and *C. difficile* share 77% ANI and should not be considered within the same genus. The rate of 16S rRNA divergence in bacteria is estimated to be 1–2% per 50 Ma (*Ochman et al., 1999*). Contradicting our ANI and core genome data, 16S rRNA sequences were highly conserved across all eight clades. This indicates that in *C. difficile* 16S rRNA gene similarity correlates poorly with measures of genomic, phenotypic, and ecological diversity, as reported in other taxa such as *Streptomyces*, *Bacillus,* and *Enterobacteriaceae* (*Janda and Abbott, 2007*; *Chevrette et al., 2019*). Another interesting observation is that C5 and the three cryptic clades had a high proportion (>90%) of MLST alleles that were absent in other clades (*Supplementary file 1e*), suggesting minimal exchange of essential housekeeping genes between these clades. Whether this reflects divergence or convergence of two species, as seen in *Campylobacter* (*Sheppard et al., 2008*), is unknown. Taken together, these data strongly support the reclassification of *C. difficile* clades C-I, C-II, and C-III as novel independent *Clostridioides* genomospecies. There have been similar genome-based reclassifications in *Bacillus* (*Liu et al., 2018*), *Fusobacterium* (*Kook et al., 2017*), and *Burkholderia* (*Loveridge et al., 2017*). Also, a recent Consensus Statement (*Murray et al., 2020*) argues that the genomics and big data era necessitate

easing of nomenclature rules to accommodate genome-based assignment of species status to non-culturable bacteria and those without 'type material', as is the case with these *Clostridioides* genomospecies.

We also found that the significant taxonomic incoherence observed in *C. difficile* was also evident in other medically important clostridia, supporting calls for taxonomic revisions (*Lawson et al., 2016*; *Oren and Rupnik, 2018*). The entire published collections of *C. perfringens*, *C. sporogenes*, and *C. botulinum* all contained sequenced strains with pairwise ANI below the 96% demarcation threshold, with 8% of *C. sporogenes* and 31% of *C. botulinum* sequenced strains below 90% ANI. These findings highlight a significant problem with the current classification of the clostridia and further demonstrate that high-resolution approaches such as whole-genome ANI can be a powerful tool for the re-classification of these bacteria (*Lawson et al., 2016*; *Oren and Rupnik, 2018*; *Murray et al., 2020*).

The NCBI SRA was dominated by C1 and C2 strains, both in number and diversity. This apparent bias reflects the research community's efforts to sequence the most prominent strains causing CDI in regions with the highest burden, for example, ST1 from humans in Europe and North America. As such, there is a paucity of sequenced strains from diverse environmental sources, animal reservoirs, or regions associated with atypical phenotypes. Cultivation bias – a historical tendency to culture, preserve, and ultimately sequence isolates that are concordant with expected phenotypic criteria – comes at the expense of 'outliers' or intermediate phenotypes. Members of the cryptic clades fit this criterion. They were first identified in 2012 but have been overlooked due to atypical toxin architecture, which may compromise diagnostic assays (discussed below). Our updated MLST phylogeny shows as many as 55 STs across the three cryptic clades (C-I, n = 25; C-II, n = 9; C-III, n = 21) (*Figure 2*). There remains a further dozen 'outliers' that could either fit within these new taxa or be the first typed representative of additional genomospecies. The growing popularity of metagenomic sequencing of animal and environmental microbiomes will certainly identify further diversity within these taxa, including nonculturable strains (*Stewart et al., 2018*; *Lu et al., 2015*).

By analysing 260 STs across eight clades, we provide the most comprehensive pangenome analysis of *C. difficile* to date. Importantly, we also show that the choice of algorithm significantly affects pangenome estimation. The *C. difficile* pangenome was determined to be open (i.e. an unlimited gene repertoire) and vast in scale (over 17,000 genes), much larger than previous estimates (~10,000 genes), which mainly considered individual clonal lineages (*Knight et al., 2019*; *Knight et al., 2016*). Conversely, comprising just 12.8% of its genetic repertoire (2232 genes), the core genome of *C. difficile* is remarkably small, consistent with earlier WGS and microarray-based studies describing ultra-low genome conservation in *C. difficile* (*Knight et al., 2015*; *Scaria et al., 2010*). Considering only C1–5, the pangenome reduced in size by 12% (2082 genes); another 519 genes were lost when considering only C1–4. These findings are consistent with our taxonomic data, suggesting that the cryptic clades, and to a lesser extent C5, contribute a significant proportion of evolutionarily divergent and unique loci to the gene pool. A large open pangenome and small core genome are synonymous with a sympatric lifestyle, characterised by cohabitation with, and extensive gene transfer between, diverse communities of prokarya and archaea (*Medini et al., 2005*). Indeed, *C. difficile* shows a highly mosaic genome comprising many phages, plasmids, and integrative and conjugative elements (*Knight et al., 2015*), and has adapted to survival in multiple niches including the mammalian gastrointestinal tract, water, soil and compost, and invertebrates (*Knight and Riley, 2019*).

Through a robust Pan-GWAS approach, we identified loci that are enriched or unique in the genomospecies. C-I strains were associated with the presence of transporter AbgB and absence of a mannose-type phosphotransferase (PTS) system. In *E. coli*, AbgAB proteins allow it to survive on exogenous sources of folate (*Carter et al., 2007*). In many enteric species, the mannose-type PTS system is essential for catabolism of fructosamines such as glucoselysine and fructoselysine, abundant components of rotting fruit and vegetable matter (*Miller et al., 2015*). C-II strains contained Zn transporter loci *znuA* and *yeiR*, in addition to Zn transporter ZupT, which is highly conserved across all eight *C. difficile* clades. *S. enterica* and *E. coli* harbour both *znuA*/*yeiR* and ZupT loci, enabling survival in Zn-depleted environments (*Sabri et al., 2009*). C-III strains were associated with major gene clusters encoding systems for ethanolamine catabolism, heavy metal transport, and spermidine uptake. The C-III *eut* gene cluster encoded six additional kinases, transporters, and transcription regulators absent from the highly conserved *eut* operon found in other clades. Ethanolamine is a valuable source of carbon and/or nitrogen for many bacteria, and *eut* gene mutations (in C1/C2) impact

toxin production *in vivo* (*Nawrocki et al., 2018*). The C-III metal transport gene cluster encoded a chelator of heavy metal ions and a multi-component transport system with specificity for iron, nickel, and glutathione. The conserved spermidine operon found in all *C. difficile* clades is thought to play an important role in various stress responses including during iron limitation (*Berges et al., 2018*). The additional, divergent spermidine transporters found in C-III were similar to regions in closely related genera *Romboutsia* and *Paeniclostridium* (data not shown). Together, these data provide preliminary insights into the biology and ecology of the genomospecies. Most differential loci identified were responsible for extra or alternate metabolic processes, some not previously reported in *C. difficile*. It is therefore tempting to speculate that the evolution of alternate biosynthesis pathways in these species reflects distinct ancestries and metabolic responses to evolving within markedly different ecological niches.

This work demonstrates the presence of toxin genes on PaLoc and CdtLoc structures in all three genomospecies, confirming their clinical relevance. Monotoxin PaLocs were characterised by the presence of *tcdR*, *tcdB,* and *tcdE*, the absence of *tcdA* and *tcdC*, and flanking by transposases and recombinases which mediate LGT (*Ramírez-Vargas and Rodriguez, 2020*; *Ramírez-Vargas et al., 2018*; *Monot et al., 2015*). These findings support the notion that the classical bi-toxin PaLoc common to clades C1–5 was derived by multiple independent acquisitions and stable fusion of monotoxin PaLocs from ancestral clostridia (*Monot et al., 2015*). Moreover, the presence of syntenic PaLoc and CdtLoc (in ST369, C-I), the latter featuring two copies of *cdtA* and *cdtR*, and a recombinase (*xerC*), further supports this PaLoc fusion hypothesis (*Monot et al., 2015*).

Bacteriophage holin and endolysin enzymes coordinate host cell lysis, phage release, and toxin secretion (*Fortier, 2018*). Monotoxin PaLocs comprising phage-derived holin (*tcdE*) and endolysin (*cwlH*) genes were first described in C-I strains (*Monot et al., 2015*). We have expanded this previous knowledge by demonstrating that syntenic *tcdE* and *cwlH* are present within monotoxin PaLocs across all three genomospecies. Moreover, since some strains contained *cwlH* but lacked toxin genes, this gene seems to be implicated in toxin acquisition. These data, along with the detection of a complete and functional (*Riedel et al., 2017*) CdtLoc contained within ΦSemix9P1 in ST343 (C-III), further substantiate the role of phages in the evolution of toxin loci in *C. difficile* and related clostridia (*Fortier, 2018*).

The CdtR and TcdR sequences of the new genomospecies are unique, and further work is needed to determine if these regulators display different mechanisms or efficiencies of toxin expression (*Chandrasekaran and Lacy, 2017*). The presence of dual copies of CdtR in ST369 (C-I) is intriguing as analogous duplications in PaLoc regulators have not been documented. One of these CdtR had a mutation at a key phosphorylation site (Asp61→Asn61) and possibly shows either reduced wild-type activity or non-functionality, as seen in ST11 (*Bilverstone et al., 2019*). This might explain the presence of a second CdtR copy.

TcdB alone can induce host innate immune and inflammatory responses leading to intestinal and systemic organ damage (*Carter et al., 2015*). Our phylogenetic analysis shows that TcdB sequences from the three genomospecies are related to TcdB in C2 members, specifically ST1 and ST41, both virulent lineages associated with international CDI outbreaks (*He et al., 2013*; *Eyre et al., 2015*), and causing classical or variant (*C. sordellii*-like) cytopathic effects, respectively (*Lanis et al., 2010*). It would be relevant to explore whether the divergent PaLoc and CdtLoc regions confer differences in biological activity, as these may present challenges for the development of effective broad-spectrum diagnostic assays, and vaccines. We have previously demonstrated that common laboratory diagnostic assays may be challenged by changes in the PaLoc of C-I strains (*Ramírez-Vargas et al., 2018*). The same might be true for monoclonal antibody-based treatments for CDI such as bezlotoxumab, known to have distinct neutralising activities against different TcdB subtypes (*Shen et al., 2020*).

Our findings highlight major incongruence in *C. difficile* taxonomy, identify differential patterns of diversity among major clades, and advance understanding of the evolution of the PaLoc and CdtLoc. While our analysis is limited solely to the genomic differences between *C. difficile* clades, our data provide a robust genetic foundation for future studies to focus on the phenotypic, ecological, and epidemiological features of these interesting groups of strains, including defining the biological consequences of clade-specific genes and pathogenic differences *in vitro* and *in vivo*. Our findings reinforce that the epidemiology of this important One Health pathogen is not fully understood. Enhanced surveillance of CDI and WGS of new and emerging strains to better inform the design of

diagnostic tests and vaccines are key steps in combating the ongoing threat posed by *C. difficile*. Last, besides *C. difficile*, we also demonstrate that a similar approach can be applied to other clostridia making a useful tool for the reclassification of these taxa.

# Materials and methods

### Key resources table

| Reagent type (species) or resource | Designation | Source or reference | Identifiers | Additional information |
|---|---|---|---|---|
| Software, algorithm | ABRicate | https://github.com/tseemann/abricate | RRID:SCR_021093 | |
| Software, algorithm | ACT: Artemis Comparison Tool | http://www.sanger.ac.uk/resources/software/act/ | RRID:SCR_004507 | |
| Software, algorithm | BactDating | https://github.com/xavierdidelot/BactDating | RRID:SCR_021092 | |
| Software, algorithm | BEAST | http://beast.bio.ed.ac.uk/ | RRID:SCR_010228 | |
| Software, algorithm | Clustal Omega | http://www.ebi.ac.uk/Tools/msa/clustalo/ | RRID:SCR_001591 | |
| Software, algorithm | Easyfig | http://easyfig.sourceforge.net/ | RRID:SCR_013169 | |
| Software, algorithm | FastANI | https://github.com/ParBLiSS/FastANI | RRID:SCR_021091 | |
| Software, algorithm | Geneious | http://www.geneious.com/ | RRID:SCR_010519 | |
| Software, algorithm | Gubbins | https://sanger-pathogens.github.io/gubbins/ | RRID:SCR_016131 | |
| Software, algorithm | iToL | https://itol.embl.de/ | RRID:SCR_018174 | |
| Other | KEGG | http://www.kegg.jp/ | RRID:SCR_012773 | Online database |
| Software, algorithm | Kraken2 | http://www.ebi.ac.uk/research/enright/software/kraken | RRID:SCR_005484 | |
| Software, algorithm | MAFFT | http://mafft.cbrc.jp/alignment/server/ | RRID:SCR_011811 | |
| Software, algorithm | MEGA | http://megasoftware.net/ | RRID:SCR_000667 | |
| Software, algorithm | MUSCLE | http://www.ebi.ac.uk/Tools/msa/muscle/ | RRID:SCR_011812 | |
| Other | NCBI RefSeq | https://www.ncbi.nlm.nih.gov/refseq/ | RRID:SCR_008420 | Online database |
| Other | NCBI Sequence Read Archive | http://www.ncbi.nlm.nih.gov/sra | RRID:SCR_004891 | Online database |
| Software, algorithm | Panaroo | https://github.com/gtonkinhill/panaroo | RRID:SCR_021090 | |
| Software, algorithm | PanGP | https://pangp.zhaopage.com/ | RRID:SCR_021089 | |
| Software, algorithm | Phandango | http://phandango.net/ | RRID:SCR_015243 | |
| Software, algorithm | Prokka | http://www.vicbioinformatics.com/software.prokka.shtml | RRID:SCR_014732 | |

*Continued on next page*

*Continued*

| Reagent type (species) or resource | Designation | Source or reference | Identifiers | Additional information |
|---|---|---|---|---|
| Other | PubMLST | http://pubmlst.org/ | RRID:SCR_012955 | Online database |
| Software, algorithm | pyani | https://pypi.org/project/pyani/ | RRID:SCR_021088 | |
| Software, algorithm | QUAST | http://bioinf.spbau.ru/quast | RRID:SCR_001228 | |
| Software, algorithm | RAxML | https://github.com/stamatak/standard-RAxML | RRID:SCR_006086 | |
| Software, algorithm | Roary | https://sanger-pathogens.github.io/Roary/ | RRID:SCR_018172 | |
| Software, algorithm | Scoary | https://github.com/AdmiralenOla/Scoary | RRID:SCR_021087 | |
| Software, algorithm | SPAdes | http://bioinf.spbau.ru/spades/ | RRID:SCR_000131 | |
| Software, algorithm | SPSS | https://www.ibm.com/products/spss-statistics | RRID:SCR_019096 | |
| Software, algorithm | SRST2 | https://github.com/katholt/srst2 | RRID:SCR_015870 | |
| Software, algorithm | TrimGalore | http://www.bioinformatics.babraham.ac.uk/projects/trim_galore/ | RRID:SCR_011847 | |

## Genome collection

We retrieved the entire collection of *C. difficile* genomes (taxid ID 1496) held at the NCBI SRA (https://www.ncbi.nlm.nih.gov/sra/). The raw dataset (as of 1 January 2020) comprised 12,621 genomes. After filtering for redundancy and Illumina paired-end data (all platforms and read lengths), 12,304 genomes (97.5%) were available for analysis.

## Multi-locus sequence typing

Sequence reads were interrogated for MLST using SRST2 v0.1.8 (*Inouye et al., 2014*). New alleles, STs, and clade assignments were verified by submission of assembled contigs to PubMLST (https://pubmlst.org/cdifficile/). A species-wide phylogeny was generated from 659 ST alleles sourced from PubMLST (dated 1 January 2020). Alleles were concatenated in frame and aligned with MAFFT v7.304. A final neighbour-joining tree was generated in MEGA v10 (*Kumar et al., 2018*) and annotated using iToL v4 [https://itol.embl.de/].

## Genome assembly and quality control

Genomes were assembled, annotated, and evaluated using a pipeline comprising TrimGalore v0.6.5, SPAdes v3.6.043, Prokka v1.14.5, and QUAST v2.344 (*Knight et al., 2019*). Next, Kraken2 v2.0.8-beta (*Wood et al., 2019*) was used to screen for contamination and assign taxonomic labels to reads and draft assemblies. Based on metadata, read depth, and assembly quality, a final dataset of 260 representative genomes of each ST present in the ENA were used for all subsequent bioinformatics analyses (C1, n = 149; C2, n = 35; C3, n = 7; C4, n = 34; C5, n = 18; C-I, n = 12; C-II, n = 3; C-III, n = 2). The list of representative genomes is available in *Supplementary file 1b*.

## Taxonomic analyses

Species-wide genetic similarity was determined by computation of whole-genome ANI for 260 STs. Both alignment-free and conventional alignment-based ANI approaches were taken, implemented in FastANI (*Jain et al., 2018*) v1.3 and the Python module pyani (*Pritchard et al., 2016*) v0.2.9, respectively. FastANI calculates ANI using a unique *k*-mer based alignment-free sequence mapping engine, whilst pyani utilises two different classical alignment ANI algorithms based on BLAST+ (ANIb) and

MUMmer (ANIm). A 96% ANI cut-off was used to define species boundaries (*Ciufo et al., 2018*). For taxonomic placement, ANI was determined for divergent *C. difficile* genomes against two datasets comprising (i) members of the *Peptostreptococcaceae* (n = 25) (*Lawson et al., 2016*) and (ii) the complete NCBI RefSeq database (n = 5895 genomes, https://www.ncbi.nlm.nih.gov/refseq/, accessed 14 January 2020). Finally, comparative identity analysis of consensus 16S rRNA sequences for *C. mangenotii* type strain DSM1289T (*Lawson et al., 2016*) (accession FR733662.1) and representatives of each *C. difficile* clade was performed using Clustal Omega https://www.ebi.ac.uk/Tools/msa/clustalo/.

## Estimates of clade and species divergence

BactDating v1.0.1 (*Didelot et al., 2018*) was applied to the recombination-corrected phylogeny produced by Gubbins (471,708 core-genome sites) with Markov chain Monte Carlo (MCMC) chains of $10^7$ iterations sampled every $10^4$ iterations with a 50% burn-in. A strict clock model was used with a rate of $2.5 \times 10^{-9}$ to $1.5 \times 10^{-8}$ substitutions per site per year, as previously defined by *He et al., 2013* and *Kumar et al., 2019*. The effective sample sizes (ESS) were >200 for all estimated parameters, and traces were inspected manually to ensure convergence. To provide an independent estimate from BactDating, BEAST v1.10.4 (*Drummond and Rambaut, 2007*) was run on a recombination-filtered gap-free alignment of 10,466 sites with MCMC chains of $5 \times 10^8$ iterations, with a $9 \times 10^{-7}$ burn-in, which were sampled every $10^4$ iterations. The strict clock model described above was used in combination with the discrete GTR gamma model of heterogeneity among sites and skyline population model. MCMC convergence was verified with Tracer v1.7.1, and ESS for all estimated parameters were >150. For ease of comparison, clade dating from both approaches was transposed onto a single MLST phylogeny. Tree files are available as *Supplementary file 3* and *4* at http://doi.org/10.6084/m9.figshare.12471461.

## Pangenome analysis

The 260 ST dataset was used for pangenome analysis with Panaroo v1.1.0 (*Tonkin-Hill et al., 2020*) and Roary v3.6.0 (*Page et al., 2015*). Panaroo was run with default thresholds for core assignment (98%) and blastP identity (95%). Roary was run with a default threshold for core assignment (99%) and two different thresholds for BlastP identity (95%, 90%). Sequence alignment of the final set of core genes (Panaroo; n = 2232 genes, 2,606,142 bp) was performed using MAFFT v7.304, and recombinative sites were filtered using Gubbins v7.304 (*Croucher et al., 2015*). A recombinant adjusted alignment of 471,708 polymorphic sites was used to create a core genome phylogeny with RAxML v8.2.12 (GTR gamma model of among-site rate-heterogeneity), which was visualised alongside pangenome data in Phandango (*Hadfield et al., 2018*). Pangenome dynamics were investigated with PanGP v1.0.1 as previously described (*Knight et al., 2019*).

Scoary (*Brynildsrud et al., 2016*) v1.6.16 was used to identify genetic loci that were statistically associated with each clade via a pan-GWAS. The Panaroo-derived pangenome (n = 17,470) was used as input for Scoary with the evolutionary clade of each genome depicted as a discrete binary trait. Scoary was run with 1000 permutation replicates, and genes were reported as significantly associated with a trait if they attained p-values (empirical, naïve, and Benjamini–Hochberg-corrected) of ≤0.05, a sensitivity and specificity of >99% and 97.5%, respectively, and were not annotated as 'hypothetical proteins'. All significantly associated genes were reannotated using Prokka and BlastP, and functional classification (KEGG orthology) was performed using the Koala suite of web-based annotation tools (*Kanehisa et al., 2016*).

## Comparative analysis of toxin gene architecture

The 260 ST genome dataset was screened for the presence of *tcdA*, *tcdB*, *cdtA*, and *cdtB* using the Virulence Factors Database (VFDB) compiled within ABRicate v1.0 (*Seemann, 2020*). Results were corroborated by screening raw reads against the VFDB using SRST2 v0.1.8 (*Inouye et al., 2014*). Both approaches employed minimum coverage and identity thresholds of 90 and 75%, respectively. Comparative analysis of PaLoc and CdtLoc architecture was performed by mapping of reads with Bowtie2 v.2.4.1 to cognate regions in reference strain R20291 (ST1, FN545816). All PaLoc and CdtLoc loci investigated showed sufficient coverage for accurate annotation and structural inference. Genome comparisons were visualised using ACT and figures prepared with Easyfig (*Ramírez-*

*Vargas et al., 2018*). MUSCLE-aligned TcdB sequences were visualised in Geneious v2020.1.2 and used to create trees in iToL v4.

## Statistical analyses

All statistical analyses were performed using SPSS v26.0 (IBM, NY). For pangenome analyses, a chi-squared test with Yate's correction was used to compare the proportion of core genes and a one-tailed Mann–Whitney U test was used to demonstrate the reduction of gene content per genome, with a p-value$\leq$0.05 considered statistically significant.

# Acknowledgements

This work was supported, in part, by funding from The Raine Medical Research Foundation (RPG002-19) and a Fellowship from the National Health and Medical Research Council (APP1138257) awarded to DRK. KI is a recipient of the Mahidol Scholarship from Mahidol University, Thailand. This work was also supported by EULac project 'Genomic Epidemiology of *Clostridium difficile* in Latin America (T020076)' and by the Millennium Science Initiative of the Ministry of Economy, Development and Tourism of Chile, grant 'Nucleus in the Biology of Intestinal Microbiota' to DPS. This research used the facilities and services of the Pawsey Supercomputing Centre (Perth, Western Australia).

# Additional information

### Competing interests

David W Eyre: DWE declares lecture fees from Gilead, outside the submitted work. The other authors declare that no competing interests exist.

### Funding

| Funder | Grant reference number | Author |
| --- | --- | --- |
| Raine Medical Research Foundation | RPG002-19 | Daniel R Knight |
| National Health and Medical Research Council | APP1138257 | Daniel R Knight |

The funders had no role in study design, data collection and interpretation, or the decision to submit the work for publication.

### Author contributions

Daniel R Knight, Conceptualization, Data curation, Formal analysis, Supervision, Funding acquisition, Validation, Investigation, Visualization, Methodology, Writing - original draft, Project administration, Writing - review and editing, Designed the study, carried out all aspects of experiments and collected the data, and analyzed data, and wrote the manuscript; Korakrit Imwattana, Conceptualization, Data curation, Formal analysis, Investigation, Visualization, Methodology, Writing - original draft, Writing - review and editing; Brian Kullin, Enzo Guerrero-Araya, Data curation, Formal analysis, Investigation, Visualization, Methodology, Writing - original draft, Writing - review and editing; Daniel Paredes-Sabja, Formal analysis, Supervision, Investigation, Methodology, Writing - original draft, Writing - review and editing; Xavier Didelot, Formal analysis, Supervision, Investigation, Visualization, Methodology, Writing - original draft, Writing - review and editing; Kate E Dingle, David W Eyre, Formal analysis, Investigation, Methodology, Writing - original draft, Writing - review and editing; César Rodríguez, Data curation, Formal analysis, Supervision, Investigation, Visualization, Methodology, Writing - original draft, Writing - review and editing; Thomas V Riley, Conceptualization, Resources, Formal analysis, Supervision, Investigation, Methodology, Writing - original draft, Project administration, Writing - review and editing

Author ORCIDs

Daniel R Knight https://orcid.org/0000-0002-9480-4733
Korakrit Imwattana http://orcid.org/0000-0002-2538-9775
Brian Kullin http://orcid.org/0000-0001-5460-1977
Xavier Didelot http://orcid.org/0000-0003-1885-500X
David W Eyre http://orcid.org/0000-0001-5095-6367
César Rodríguez http://orcid.org/0000-0001-5599-0652
Thomas V Riley https://orcid.org/0000-0002-1351-3740

Decision letter and Author response
Decision letter https://doi.org/10.7554/eLife.64325.sa1
Author response https://doi.org/10.7554/eLife.64325.sa2

## Additional files

### Supplementary files

• Supplementary file 1. Summary data: multi-locus sequence type, pangenome, taxonomy, genome-wide association study, toxins.

• Supplementary file 2. Tree file, global multi-locus sequence type.

• Supplementary file 3. Tree file, BEAST.

• Supplementary file 4. Tree file, BactDating.

• Transparent reporting form

### Data availability

All data generated or analysed during this study are included in the manuscript and Supplementary Data which is hosted at figshare http://doi.org/10.6084/m9.figshare.12471461. Data files on figshare include: [1] Full MLST data for all 12000+ *C. difficile* genomes (Figure 1). [2] Whole-genome ANI analyses (Table 1, Figure 3, Figure 5). [3] Tree files for phylogenetic analyses (Figure 2, Figure 4). [4] Pangenome data (Figure 6). [5] Pan-GWAS data (Table 2). [6] Comparative genomic analysis of virulence gene architecture (Figure 7). We retrieved the entire collection of *C. difficile* genomes (taxid ID 1496) held at the NCBI SRA [https://www.ncbi.nlm.nih.gov/sra/]. The raw dataset (as of 1st January 2020) comprised 12,621 genomes. These genomes comprise hundreds, maybe thousands of publications. The individual accession numbers for all genomes analysed in this study are provided in the Supplementary Data at http://doi.org/10.6084/m9.figshare.12471461.

The following dataset was generated:

| Author(s) | Year | Dataset title | Dataset URL | Database and Identifier |
|---|---|---|---|---|
| Knight DR, Imwattana K, Kullin B, Guerrero-Araya E, Paredes-Sabja D, Didelot X, Dingle KE, Eyre DW, Rodríguez Cs, Riley TV | 2021 | Supplementary Data: Summary data on multi-locus sequence type, pangenome, taxonomy, genome-wide association study, toxins. Also, raw tree files for phylogenetic analyses. | http://doi.org/10.6084/m9.figshare.12471461 | figshare, 10.6084/m9.figshare.12471461 |

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
