## [Decision Letter]

**Acceptance summary:**

The work presented by Knight et al. in "Major genetic discontinuity and novel toxigenic species in *Clostridioides difficile* taxonomy" provides a thorough and robust examination of publicly available C. difficile genomes to deliver a much-needed update of *C. difficile* phylogeny, in particular the cryptic clades of *C. difficile*. The analyses will be useful for clinical bacterial taxonomy and advance understanding of the evolution of these important species

**Decision letter after peer review:**

[Editors’ note: the authors submitted for reconsideration following the decision after peer review. What follows is the decision letter after the first round of review.]

Thank you for submitting your work entitled "Major genetic discontinuity and novel toxigenic species in *Clostridioides difficile* taxonomy" for consideration by *eLife*. Your article has been reviewed by 2 peer reviewers, and the evaluation has been overseen by a Reviewing Editor and a Senior Editor. The following individual involved in review of your submission has agreed to reveal their identity: Steven Mileto (Reviewer #1).

Our decision has been reached after consultation between the reviewers. Based on these discussions and the individual reviews below, we regret to inform you that your work will not be considered further for publication in *eLife*.

We appreciate this study and find that the conclusions that reclassify Clostridiodes are largely justified by the data/analysis. The major concern is that the work represents the application of standard approaches to refine species classification, as opposed to either proposing a novel approach to classify species or defining a split that might be more surprising and/or clinically significant (e.g. Kumar et al. Nature Genetics, 2019). Consequently, despite being a useful contribution to the literature we believe it is more suitable for a specialized journal.

*Reviewer #1:*

The work presented by Knight et al. in "Major genetic discontinuity and novel toxigenic species in Clostridioides difficile taxonomy" is of excellent quality and spans several of the themes of *eLife*. The manuscript provides a thorough and robust examination of publicly available C. difficile genomes, to deliver a much-needed update of C. difficile phylogeny, in particular the cryptic clades of C. difficile. However, there are some further clarifications could be included to confirm if the cryptic clades of C. difficile, and the 26 unclassified STs (which seemingly form 4 distinct clusters) should indeed be assigned to the Clostridioides genus, distinct from both C. mangenotii and C. difficile. Thus, I would accept this manuscript, without additional work, but with minor amendments.

Lines 96-97 and Figure 2: Figure 2 suggests the 26 unclassified STs form at least 4 distinct clusters, yet these STs are classified as outliers. Could you please comment on why these are considered outliers? Or do these STs represent new cryptic clades? C-IV, C-V etc.? And do these unclassified STs also fit in to the criteria for the novel independent Clostridioides genomospecies?

Lines 161-162; Table 1: C. mangenotii is referred to as Clostridioides mangenotii on lines 161-162, but has been listed as Clostridium mangenotii in table 1. Was this intentional? Or should this be Clostridioides mangenotii as C. difficile is also listed as Clostridioides difficile?

Figure 6: Many of the numbers and symbols on the figure are difficult to see e.g. Figure 6A the values listed above each data point are extremely small. Can these values/symbols be increased?

Lines 224-225: Given that C. difficile strains lacking tcdA and tcdB can still cause infections, consider rephrasing "indicating their ability to cause CDI".

Figure 7: As with Figure 6, many of the numbers and symbols on the figure are difficult to see. Can these values/symbols be increased?

Were the unclassified STs included in the species wide ANI analyses in Figure 3? If similar analyses were performed for these STs and given the clusters that are presented in Figure 2 would this support the idea that they may also fit in to the criteria for the novel independent Clostridioides genomospecies?

Similarly, were these same unclassified STs included in the BactDating and BEAST analyses? Or the pairwise ANI and 16S rRNA value comparisons in Figure 5? Or the pangenome and toxin gene analysis also presented in Figures 6 and 7? And would this add further strength to the idea that these "outliers" could be the first typed representatives of additional genomospecies?

Lastly, your conclusions are a little too on the fence. You have presented sufficient evidence to suggest that the cryptic clades of C. difficile likely represent novel independent Clostridioides genomospecies, but dilute out the importance of this throughout the discussion and conclusions. Although controversial, the evidence provided gives credence to these claims, and the text should be changed to reflect this.

*Reviewer #2:*

In this manuscript, Knight et al. examine the genetic diversity in >12,000 publicly available C. difficile genomes in order to characterize genomic evidence of taxonomic incoherence among this genomically diverse pathogen. Their primary analysis employs average nucleotide identity thresholds to identify species boundaries, with secondary analyses examining core genome size changes, gene content, and estimated emergence dates. The authors' main conclusion is that the previously identified C. difficile cryptic clades CI-III are genomically divergent enough from the main clades C1-5 to warrant classification as different genomospecies. This paper is a useful contribution in benchmarking our understanding of the genetic diversity of C. difficile using all currently publicly available genomes, but the results are largely unsurprising given previous phylogenetic analyses involving clades 1-5 and CI-III, and is therefore probably best suited for a specialty journal. Additionally, in some instances, the methods lack details, reducing their interpretability and reproducibility.

1. There are some claims that are too strong and not supported by the data or literature, including the claim that the rise of community-associated CDI is likely due to presence of C. difficile in livestock (Lines 53-54 – far too little evidence to make such a sweeping claim), the statement of apparent rapid population expansion into clades C1-4 (Lines 278-279 – only shown for certain sequence types and greatly impacted by observation bias), the statement that these findings "impacts the diagnosis of CDI worldwide" (Lines 37-38 -too grandiose given limited evidence of the clinical importance of the cryptic clades).

2. Generally, it is hard to discern which sets of genomes and variants were used for each of the bioinformatic analyses that are described. If there are a limited number of genome sets it might be useful to define them in the results to allow the reader to more easily follow along and understand the scope of different analyses.

3. The dated phylogenomic analyses methods would benefit from a more thorough assessment of model assumptions along with more description of the sources of bias and uncertainty at play. Specific questions are:

– Was temporal signal in the data evaluated?

– What are the potential impacts of using a single clock model and demographic prior for such a diverse set of taxa?

– Was the clock rate restricted to the cited 2.5x10-9 – 1.5 x 10-8 range? What clock prior distribution was applied?

– Were relaxed clock priors explored?

– What went into the selection of the demographic model prior in BEAST? Were alternative models evaluated?

– The significant uncertainty in the divergence estimates should be emphasized/listed as a limitation.

4. Similarly, the pangenome analyses could be more thoroughly described, and the relevance of the core-genome size changes more robustly explored. Specifically:

– How did the core genome change when excluding any of C1-5? Were these changes much different than when excluding CI-III?

– The differences between Roary and Panaroo are notable, and potentially important for the microbial genomics community. More details should be provided on these results and how sensitive they are to the input parameters of the respective programs (e.g. collapsing paralogs in Roary and percent identity for orthologs). In addition, it is important to know if any filtering was done with respect to the quality of assemblies, which could have a significant impact on Roary's behavior.

[Editors’ note: further revisions were suggested prior to acceptance, as described]

Thank you for submitting your article "Major genetic discontinuity and novel toxigenic species in *Clostridioides difficile* taxonomy" for consideration by *eLife*. Your article has been reviewed by 0 peer reviewers, and the evaluation has been overseen by a Reviewing Editor and Dominique Soldati-Favre as the Senior Editor.

The reviewers have discussed your revision with one another, and the Reviewing Editor has drafted this to help you prepare a revised submission.

Summary:

The work presented by Knight et al. in "Major genetic discontinuity and novel toxigenic species in Clostridioides difficile taxonomy" is of excellent quality and spans several of the themes of *eLife*. The manuscript provides a thorough and robust examination of publicly available C. difficile genomes, to deliver a much-needed update of C. difficile phylogeny, in particular the cryptic clades of C. difficile.

Essential Revisions:

1) The only remaining technical critique is with the dated phylogenetic analyses. The authors would be well served to not hang their hats on the point estimates of clade divergences, which have massive confidence intervals that are almost completely non-overlapping between the two methods applied. Please also remove implications that these dated analyses are robust (lines 316-317), as the lack of agreement between methods and wide confidence intervals do not support this assertion. Rather, what is robust is the branching order of the clades, which is supported by different comparative genomic and phylogenomic analyses, so I would suggest focusing more on this.

---

## [Author Response]

[Editors’ note: The authors appealed the original decision. What follows is the authors’ response to the first round of review.]

We appreciate this study and find that the conclusions that reclassify Clostridiodes are largely justified by the data/analysis. The major concern is that the work represents the application of standard approaches to refine species classification, as opposed to either proposing a novel approach to classify species or defining a split that might be more surprising and/or clinically significant (e.g. Kumar et al. Nature Genetics, 2019). Consequently, despite being a useful contribution to the literature we believe it is more suitable for a specialized journal.Reviewer #1:The work presented by Knight et al. in "Major genetic discontinuity and novel toxigenic species in Clostridioides difficile taxonomy" is of excellent quality and spans several of the themes of eLife. The manuscript provides a thorough and robust examination of publicly available C. difficile genomes, to deliver a much-needed update of C. difficile phylogeny, in particular the cryptic clades of C. difficile. However, there are some further clarifications could be included to confirm if the cryptic clades of C. difficile, and the 26 unclassified STs (which seemingly form 4 distinct clusters) should indeed be assigned to the Clostridioides genus, distinct from both C. mangenotii and C. difficile. Thus, I would accept this manuscript, without additional work, but with minor amendments.Lines 96-97 and Figure 2: Figure 2 suggests the 26 unclassified STs form at least 4 distinct clusters, yet these STs are classified as outliers. Could you please comment on why these are considered outliers? Or do these STs represent new cryptic clades? C-IV, C-V etc.? And do these unclassified STs also fit in to the criteria for the novel independent Clostridioides genomospecies?

We took a two-step approach to *C. difficile* clade assignment and genome analysis. First, working with curators at PubMLST (Prof Martin Maiden and Dr Kate Dingle), we screened the 12k genomes from the ENA for MLST and were able to confidently assign 272 STs to 8 described clades. This represented 40% of the known STs in the PubMLST database (Figure 1).

Second, based on PubMLST data and bootstraps values of 1.0 in all monophyletic nodes of the cryptic clades (Figure 2), we could confidently assign 25, 9 and 10 STs to cryptic clades I, II and III, respectively. There remained 26 STs spread across the phylogeny that did not fit within a specific clade (defined as outliers). These 26 STs were not found in our 12k WGS dataset so are either Sanger sequenced strains or unpublished WGS data. Consequently, they were not included in the genome analysis.

However, we agree with the reviewer that some of these outliers do appear to form clusters and likely represent additional novel *Clostridioides* genomospecies. These certainly warrant further study when sufficient WGS sequence data is available.

Lines 161-162; Table 1: C. mangenotii is referred to as Clostridioides mangenotii on lines 161-162, but has been listed as Clostridium mangenotii in table 1. Was this intentional? Or should this be Clostridioides mangenotii as C. difficile is also listed as Clostridioides difficile?

*Clostridioidesmangenotii* is the only other known member of *Clostridioides*. Table 2 has been updated accordingly.

Figure 6: Many of the numbers and symbols on the figure are difficult to see e.g. Figure 6A the values listed above each data point are extremely small. Can these values/symbols be increased? Figure 7: As with Figure 6, many of the numbers and symbols on the figure are difficult to see. Can these values/symbols be increased?

All manuscript figures are standardised for font/symbol size and were designed for optimal viewing online. If accepted, and the production team feels appropriate, we could revise the font size.

Lines 224-225: Given that C. difficile strains lacking tcdA and tcdB can still cause infections, consider rephrasing "indicating their ability to cause CDI".

The text has been revised, see lines 241-243.

Were the unclassified STs included in the species wide ANI analyses in Figure 3? If similar analyses were performed for these STs and given the clusters that are presented in Figure 2 would this support the idea that they may also fit in to the criteria for the novel independent Clostridioides genomospecies?Similarly, were these same unclassified STs included in the BactDating and BEAST analyses? Or the pairwise ANI and 16S rRNA value comparisons in Figure 5? Or the pangenome and toxin gene analysis also presented in Figures 6 and 7? And would this add further strength to the idea that these "outliers" could be the first typed representatives of additional genomospecies?

We agree that these outliers likely represent additional genomospecies, and certainly warrant further study when sufficient WGS sequence data is available. For this study, only STs definitively belonging to cryptic clades CI-III were included in the detailed bioinformatics analyses e.g., ANI, BEAST/ BactDating, 16S rRNA, pangenome, and toxin gene analysis (Figure 3-7). As this point was also raised by Reviewer 2, we have updated the text to clearly delineate those 260 strains included in the genomic analyses, see lines 97-101 and 462-466. The list of representative genomes is now available in Supplementary File 1.

Lastly, your conclusions are a little too on the fence. You have presented sufficient evidence to suggest that the cryptic clades of C. difficile likely represent novel independent Clostridioides genomospecies, but dilute out the importance of this throughout the discussion and conclusions. Although controversial, the evidence provided gives credence to these claims, and the text should be changed to reflect this.

While we agree with the sentiment, we think we should remain a little more measured. Although we provide strong evidence of genomic incongruence, further studies are still needed to confirm the phenotypic, ecological and epidemiological features of these genomospecies. We have added a paragraph at the end of the discussion to acknowledge this limitation, see lines 437-444.

Reviewer #2:In this manuscript, Knight et al. examine the genetic diversity in >12,000 publicly available C. difficile genomes in order to characterize genomic evidence of taxonomic incoherence among this genomically diverse pathogen. Their primary analysis employs average nucleotide identity thresholds to identify species boundaries, with secondary analyses examining core genome size changes, gene content, and estimated emergence dates. The authors' main conclusion is that the previously identified C. difficile cryptic clades CI-III are genomically divergent enough from the main clades C1-5 to warrant classification as different genomospecies. This paper is a useful contribution in benchmarking our understanding of the genetic diversity of C. difficile using all currently publicly available genomes, but the results are largely unsurprising given previous phylogenetic analyses involving clades 1-5 and CI-III, and is therefore probably best suited for a specialty journal. Additionally, in some instances, the methods lack details, reducing their interpretability and reproducibility.

The editors refer to the paper by Kumar *et al.* 2019 (Nat Genet 51:1315-1320) and note a major concern that the work represents the application of standard approaches. The Kumar paper (which co-author Riley contributed to) focused on only 906 strains of only clades 1-5 and did not include any genomes from the cryptic clades, nor focus on pathogenicity, or employ GWAS or pangenome analysis to investigate speciation in *C. difficile*. Our new paper is the first species-wide analysis of *C. difficile* taxonomy, and the largest genomic study of *C. difficile* to date encompassing over 12k genomes from strains collected worldwide. WGS and ANI analysis is state of the art and supersedes the wet lab hybridisation method used for many decades. This approach has now been adopted by the NCBI as the gold standard for taxonomy, see Ciufo *et al.* Int J Syst Evol Microbiol 68, 2386-2392; Jain *et al.* 2018 Nat Commun 9, 5114; and Richter et al. 2019 Proc Natl Acad Sci U S A 106, 19126-19131.

Reviewer 2 felt our results are largely unsurprising given previous phylogenetic analyses. Previous work based on MLST (0.08% of a genome) indicated that the cryptic clades were highly divergent (Dingle KE, et al. 2014 Genome Biol Evol 6, 36-52 and Didelot X, et al. 2012 Genome Biol 13, R118), however, until our study, whether they truly belonged to *C. difficile* or represented novel species was unknown. We used high-throughput genomic techniques (ANI, Bayesian dating, GWAS and pangenomic analysis) to demonstrate that this is indeed the case. Our study rewrites the global population structure of *C. difficile* specifically and the taxonomy of the *Peptostreptococcaceae* in general. We show that these cryptic strains harbour unique and highly divergent toxin gene architecture. These are novel and important findings that not only provide insights into the evolution of pathogenicity in *C. difficile*, but they will impact CDI diagnosis, as toxin diversity is critical for molecular-based diagnostics for CDI.

We address the need for additional methods details in the responses below.

1. There are some claims that are too strong and not supported by the data or literature, including the claim that the rise of community-associated CDI is likely due to presence of C. difficile in livestock (Lines 53-54 – far too little evidence to make such a sweeping claim), the statement of apparent rapid population expansion into clades C1-4 (Lines 278-279 – only shown for certain sequence types and greatly impacted by observation bias), the statement that these findings "impacts the diagnosis of CDI worldwide" (Lines 37-38 -too grandiose given limited evidence of the clinical importance of the cryptic clades).

Regarding points above:

i) We have revised the text, simply highlighting the well-documented link between CA-CDI and *C. difficile* sources in animals and the environment, see lines 51-52.

ii) We stand by this observation. It is supported by our robust Bayesian analyses (Figure 4), which show that the last common ancestor of clades 1-4 existed around 1.61 mya, thus the population expansion of these clades occurred rapidly (relative to the divergence of clades 5 and cryptic clades). Moreover, we show that this divergence is consistent across >200 STs in clades 1-4, not just certain epidemic sequence types as we suspect the reviewer is referring to.

iii) It is true that the clinical importance of the cryptic clades is still unclear, however, as taxonomy underpins accurate CDI diagnosis, our findings are likely to affect/influence CDI diagnosis. We have slightly revised the abstract, see lines 36-37.

2. Generally, it is hard to discern which sets of genomes and variants were used for each of the bioinformatic analyses that are described. If there are a limited number of genome sets it might be useful to define them in the results to allow the reader to more easily follow along and understand the scope of different analyses.

We agree that the manuscript does not clearly define the sets of genomes used for the bioinformatics analyses. We have revised the text in both the results and methods sections, see lines 97-101 and 462-466. The list of representative genomes is now available in Supplementary File 1.

3. The dated phylogenomic analyses methods would benefit from a more thorough assessment of model assumptions along with more description of the sources of bias and uncertainty at play. Specific questions are:– Was temporal signal in the data evaluated?

Yes, we did try this in both BactDating and BEAST, and with both software independently we came to the same conclusion that the temporal signal is not strong enough to estimate the parameters of the clock model. This is not surprising, because the temporal signal is a function of the range of dates used in sampling divided by the overall TMRCA and, in our case, we know that the former is a very small fraction of the latter.

– What are the potential impacts of using a single clock model and demographic prior for such a diverse set of taxa?

Our approach to dating is not ideal, but it is the only thing that can be done since the data is not informative about the clock model as mentioned above. The demographic prior is unlikely to be an issue though since there are so many mutations on the branches of the tree that any prior would be overwhelmed by the likelihood. Moreover, we evaluated multiple molecular clock estimates (covering almost an order of magnitude) and ultimately used the same fixed clock model as Kumar *et al.* 2019 (Nat Genet 51:1315-1320)

– Was the clock rate restricted to the cited 2.5x10-9 – 1.5 x 10-8 range? What clock prior distribution was applied?

In BactDating the analysis was performed twice, once with the low rate and once with the high rate (from Kumar et al.) and the results were combined to show the full range of possible dates under these two extremes.

– Were relaxed clock priors explored?

No, since there is no information about the mean rate there is no information about how this mean varies over lineages. But the fact that we used a range of rates covering almost an order of magnitude means that the estimated dating is equivalent to having a relaxed clock with the same range.

– What went into the selection of the demographic model prior in BEAST? Were alternative models evaluated?

As mentioned above, the prior of the tree is not going to play any role at all when we have so many mutations on the tree.

– The significant uncertainty in the divergence estimates should be emphasized/listed as a limitation.

We agree and have revised the text, see lines 315-319.

4. Similarly, the pangenome analyses could be more thoroughly described, and the relevance of the core-genome size changes more robustly explored. Specifically:– How did the core genome change when excluding any of C1-5? Were these changes much different than when excluding CI-III?

The impact on core genome size after removing groups of clades (1-5 only and 1-4 only) are plainly illustrated in Figure 6 and detailed in lines 203-205 e.g. “The size of the pangenome reduced by 2,082 genes with the exclusion of clades CI-III, and a further 519 genes with the exclusion of C5.”

– The differences between Roary and Panaroo are notable, and potentially important for the microbial genomics community. More details should be provided on these results and how sensitive they are to the input parameters of the respective programs (e.g. collapsing paralogs in Roary and percent identity for orthologs). In addition, it is important to know if any filtering was done with respect to the quality of assemblies, which could have a significant impact on Roary's behavior.

We agree with the reviewer that the significant differences seen in the analysis with Roary and Panaroo will be of importance for the microbial genomics community in general. We now provide some example of the effect of the change in parameters in the outcomes of Panaroo and Roary, see lines 205-209 Overall, Roary tends to overestimate the number of pangenome in all settings (ranging from ~25% to almost 90%). This is likely because Panaroo has an additional step where the program polishes the annotated genomes before the analysis. This is explained in line 209-213.

[Editors’ note: what follows is the authors’ response to the second round of review.]

Essential Revisions:1) The only remaining technical critique is with the dated phylogenetic analyses. The authors would be well served to not hang their hats on the point estimates of clade divergences, which have massive confidence intervals that are almost completely non-overlapping between the two methods applied. Please also remove implications that these dated analyses are robust (lines 316-317), as the lack of agreement between methods and wide confidence intervals do not support this assertion. Rather, what is robust is the branching order of the clades, which is supported by different comparative genomic and phylogenomic analyses, so I would suggest focusing more on this.

We are happy to agree with the points made regarding uncertainty in divergence estimates and have revised the manuscript in three areas, ensuring there is greater emphasis on presenting emergence dates as ranges (confidence/credible intervals) rather than single point estimates (medians) as before. We have also provided some additional discussion around the limitations of our approach.

The revised parts of the manuscript are lines 132-140 and 153-165 (Results), and lines 317-352 (Discussion). There are also updates to the RRID information in the Key Resources Table.